# From Attention to Diffusion: A Unified Entropic Optimal Transport View

## Abstract

We show that transformer attention and diffusion models are discretizations of the same entropy-regularized optimal transport (OT) flow. A single attention layer is a KL-proximal (JKO/mirror) step in an OT potential; stacking layers yields probability paths that converge to a probability–flow ODE (PF–ODE) on the simplex. Our construction uses a causal, semi-relaxed EOT that preserves attention masking while retaining OT geometry. We derive a finite-depth error bound controlled by a budget $\Xi_L$ (quantifying continuum validity) and prove that stacked attention weakly approximates time-inhomogeneous, anisotropic reverse diffusions with an error that separates time discretization, logit variation, and optional degeneracy regularization. Geometrically, we characterize exact Schrödinger Bridge (SB) alignment via a rotational energy $\mathcal{R}$ that vanishes if and only if the path is SB, and serves as a practical diagnostic otherwise. The framework yields testable predictions: (i) the continuum approximation is accurate when $\Xi_L$ is small; (ii) depth exhibits diminishing returns beyond a threshold set by contraction and step size; and (iii) lower $\mathcal{R}$ correlates with improved generations. We validate these predictions with a diagnostic suite (P0–P4): BV/continuity gating (with abstention on failure), PF–ODE adequacy, curvature/locking geometry, and SB energy. Evidence spans two tracks—Transformers (core diagnostics) and a compact image diffusion model (parity and first-order weak-error behavior)—with validity conditions and diagnostic abstention protocols detailed in Appendix N.4. These insights motivate mobility-aware temperature scheduling and certified early exit, conserving depth while preserving transport geometry.

## 1 Introduction

Transformers and diffusion models appear fundamentally different, yet we show they instantiate two discretizations of the same entropy-regularized optimal transport flow. One attention layer performs a KL-proximal step in an optimal transport potential, and depth plays the role of time for the induced probability dynamics on the simplex.

This unification builds on and extends several research streams that have developed in isolation. Attention mechanisms have been interpreted through optimal transport in recent work (Sander et al., 2022; Tay et al., 2020; Xu et al., 2023; Daneshmand, 2024), but prior approaches typically employ balanced optimal transport formulations that are fundamentally incompatible with causal masking in autoregressive language modeling. We resolve this by proving that standard row-softmax attention precisely solves a semi-relaxed entropic optimal transport problem that preserves autoregressive causality. Continuous-depth interpretations of neural networks through neural ordinary differential equations (Chen et al., 2018; Dupont et al., 2019; Bai et al., 2019) have been extended to transformers (Zhang et al., 2021; Chen et al., 2023; Kan et al., 2025), but existing analyses typically lack rigorous finite-depth error control. We strengthen these perspectives by introducing an explicit bounded variation regime with quantitative finite-depth error bounds controlled by a budget parameter, proving that the continuous-depth limit satisfies a well-posed probability flow ordinary differential equation on the probability simplex with the softmax Jacobian acting as a mobility tensor.

Score-based generative models have revealed fundamental connections between stochastic differential equations and deterministic probability flow ordinary differential equations (Song et al., 2021; Huang et al., 2021; Lipman et al., 2022). We demonstrate that transformer attention implements discretizations of probability flow ordinary differential equations in the same geometric family as those underlying diffusion models but operating under semi-relaxed entropic regularization, explaining why autoregressive-diffusion hybrid architectures work well empirically (Hoogeboom et al., 2022; Ma et al., 2025). Schrödinger Bridge theory provides a dynamic formulation of entropy-regularized optimal transport (Léonard, 2014; De Bortoli et al., 2021; Shi et al., 2023); we operationalize Schrödinger Bridge alignment within transformer attention dynamics by defining a rotational energy quantity that measures deviations from optimality and vanishes if and only if the attention-induced flow satisfies the Schrödinger Bridge characterization, transforming abstract optimality conditions into practical diagnostics estimable from model activations. Comprehensive literature review with detailed comparisons appears in Appendix C.

**Contributions.** Under mild regularity assumptions (detailed in Section 2.2), our main results are:

1. *Layer-level principle.* Standard row-softmax attention implements a principled Kullback-Leibler proximal transport step in the sense of mirror descent or Jordan-Kinderlehrer-Otto schemes, establishing the foundational connection between neural architecture and optimal transport geometry formalized in Proposition 2.1.
2. *Depth-to-time convergence with explicit rates.* The discrepancy between discrete layer dynamics and continuous probability flow is controlled by a finite-depth budget parameter denoted $\Xi_L$ and defined precisely through bounded variation conditions in Theorem 3.1, providing quantitative error bounds that determine when continuum approximations apply to finite architectures.
3. *Diffusion unification through weak approximation.* Stacked attention layers weakly approximate time-inhomogeneous anisotropic reverse diffusions in probability law, with error that separates discretization effects from logit variation and optional degeneracy regularization, as established through the anisotropic Fokker-Planck analysis in Theorem 4.5.
4. *Schrödinger Bridge alignment certificate.* A rotational energy quantity denoted $\mathcal{R}$ provides a necessary and sufficient condition for exact Schrödinger Bridge alignment, quantifying deviations from gradient flow structure and serving as a practical diagnostic for transport optimality as formalized in Theorem 5.2.

The framework yields three falsifiable predictions: continuum approximation accuracy when $\Xi_L$ is small, diminishing returns from depth when mobility degrades, and correlation between low $\mathcal{R}$ and improved generation quality. Our empirical study tests these predictions across transformer language models and compact image diffusion, with extended protocols in Appendix 3–1. Figure 5 visualizes how these results connect: Proposition 2.1 establishes the foundational single-layer principle; Theorem 3.1 extends this to finite-depth convergence through bounded variation compactness; Theorem 3.7 provides well-posedness infrastructure supporting Theorem 4.5's diffusion unification; and Theorem 5.2 characterizes transport optimality via rotational energy. The diagram identifies the functional analysis machinery underlying each result, directly addressing proof architecture questions.

## 2 Preliminaries and Conceptual Framework

### 2.1 Conceptual Overview

Before establishing formal machinery, we outline the key geometric quantities. The softmax Jacobian $J_{\mathrm{sm}}(z)$ acts as the mobility tensor on the probability simplex, with temperature modulating transport capacity via $J_{\mathrm{sm}}^{\tau}(z) = \tau^{-1} J_{\mathrm{sm}}(z/\tau)$. The finite-depth budget $\Xi_L$ quantifies how well discrete layers approximate continuous flow; small $\Xi_L$ ensures the probability-flow ordinary differential equation accurately captures layerwise behavior. Rotational energy $\mathcal{R}$ measures deviation from optimal transport; exact Schrödinger Bridge alignment occurs when $\mathcal{R}$ vanishes.

## 2.2 Mathematical Preliminaries and Notation

This subsection establishes notation, states the standing assumptions used throughout, and records the layer-level optimal transport view we invoke in subsequent analysis.

**Global Assumptions.** We collect here the global assumptions used throughout. Assumptions 2.1 and 3.1 provide the bounded-variation and regularity conditions, with the latter adding architectural consistency for the continuum-limit results.

**Assumption 2.1** (Bounded variation and architectural consistency)**.** *We work on compact subsets where all quantities are well-defined. Unless stated otherwise, we assume:*

1. *Bounded-variation logits with uniform mean: Let $z^{(\ell)}$ denote the layer logits and $\Delta z^{(\ell)} := z^{(\ell+1)} - z^{(\ell)}$. We assume that the averaged per-layer logit variation*

$$C_{\mathrm{BV}} := \frac{1}{L} \sum_{\ell=0}^{L-1} \|\Delta z^{(\ell)}\|_\infty$$

*remains bounded by a constant $C_{\mathrm{BV}}^\star$ uniformly across all layer counts $L$. Equivalently, the total variation $\sum_{\ell=0}^{L-1} \|\Delta z^{(\ell)}\|_\infty$ grows at most linearly in $L$, so that typical layer-to-layer changes, rather than accumulated variation, control the quality of continuum approximation.*

2. *Local drift regularity. The effective drift $b(\cdot, t)$ is locally Lipschitz in its state argument on bounded sets with Lipschitz constant $L_b$ and is locally bounded by $M_b$.*

3. *Mobility bounds. For $p = \mathrm{softmax}(z/\tau)$ with temperature $\tau > 0$, the Jacobian $J_{\mathrm{sm}}(z) = \mathrm{Diag}(p) - pp^\top$ satisfies operator-norm and derivative bounds on the relevant compact domain; denote $\Lambda_J := \sup \|J_{\mathrm{sm}}(z)\|_{\mathrm{op}}$ and $L_J := \sup \|\nabla J_{\mathrm{sm}}(z)\|_{\mathrm{op}}$.*

4. *Simplex invariance. Probability vectors $p$ remain in the simplex under the dynamics considered; faces are handled by the standard tangent-space restriction.*

**Softmax and Mobility.** Given logits $z \in \mathbb{R}^V$ and temperature $\tau > 0$, the softmax operation and its induced mobility tensor are defined by

$$p = \mathrm{softmax}(z/\tau), \qquad p_i = \frac{\exp(z_i/\tau)}{\sum_j \exp(z_j/\tau)}, \qquad J_{\mathrm{sm}}(z) = \mathrm{Diag}(p) - pp^\top.$$

The Jacobian $J_{\mathrm{sm}}(z)$ characterizes how probability mass flows under logit perturbations, acting as the mobility tensor that governs transport dynamics on the probability simplex.

**Remark (Sharp Mobility Bound).** We have $\|J_{\mathrm{sm}}(z)\|_{\mathrm{op}} \leq \frac{1}{2\tau}$, with equality at distributions $p = (\frac{1}{2}, \frac{1}{2}, 0, \ldots, 0)$. In particular, for $\tau = 1$, $\|J_{\mathrm{sm}}(z)\|_{\mathrm{op}} \leq \frac{1}{2}$ and the spectrum lies in $[0, \frac{1}{2}]$, collapsing to $\{0\}$ as $\max_i p_i \to 1$. A proof is provided in Appendix B. This sharp bound explains why temperature scheduling proves essential for maintaining mobility in deep networks as distributions become increasingly peaked.

## 2.3 Semi-relaxed entropic optimal transport for attention

Standard attention with row-softmax normalization solves a semi-relaxed entropic optimal transport problem that preserves autoregressive causality. For a query vector $q \in \mathbb{R}^{d_k}$ and key vectors $k_j \in \mathbb{R}^{d_k}$, define the cost $c_j = -q \cdot k_j$, so that high similarity corresponds to low transport cost. Given a reference distribution $u \in \Delta^{V-1}$ (typically uniform) and temperature $\tau > 0$, the semi-relaxed entropic OT problem is

$$\min_{p \in \Delta^{V-1}} \left\{ \sum_{j=1}^V p_j c_j + \tau \, \mathrm{KL}(p\|u) \right\}. \tag{1}$$

The first-order optimality conditions yield the softmax solution

$$p_j = \frac{\exp(q \cdot k_j/\tau)}{\sum_{k=1}^V \exp(q \cdot k_k/\tau)} = \mathrm{softmax}(qK^\top/\tau)_j,$$

so that each attention row solves a semi-relaxed entropic OT problem where the row-stochastic constraint is enforced but column marginals are unconstrained, preserving autoregressive structure. Causal masking is implemented by assigning infinite cost $c_j = +\infty$ to masked positions. The complete derivation including the Lagrangian formulation, uniqueness, and masked formulation appears in Appendix B.

**Proposition 2.1** (Attention as KL-Proximal/JKO Step). *Let $c_j = -q \cdot k_j$ and $\tau > 0$. For any full-support reference $u$,*

$$p^+ \in \arg\min_{p \in \Delta} \left\{ \langle c, p \rangle + \tau \, \mathrm{KL}(p \, \| \, u) \right\}.$$

*Stacking such updates discretizes a Kullback-Leibler mirror descent or Jordan-Kinderlehrer-Otto flow under the assumptions in Section 2.2. The proof is given in Appendix B.*

This proposition establishes that each attention layer implements a principled optimal transport step, providing the foundation for our continuous-depth analysis in subsequent sections.

# 3 Discrete Continuity and the Continuous-Depth Limit

## 3.1 Bounded variation regime and practical implications

The transition from discrete layers to continuous dynamics requires controlling the accumulation of changes across depth. We formalize this through a bounded-variation (BV) condition that captures when transformers exhibit smooth evolution rather than abrupt transitions.

**Assumption 3.1** (Bounded variation and architectural consistency). *Let $\delta t = 1/L$ and $t_\ell = \ell/L$. We assume:*

1. *Bounded total variation: $\sum_\ell \|\Delta z^{(\ell)}\|_2 \leq C$ (uniformly in $L$).*
2. *Uniform boundedness (tightness): $\sup_\ell \|z^{(\ell)}\|_2 \leq C_z$.*
3. *Architectural consistency (identification): local-regression estimates $\hat{b}_L$ converge to $b$ on compacts; see Appendix D.*

*Note. Weak $L^1$ convergence of $D_L$ to $b$ is not assumed here; it follows from Lemma 3.5 via the calibration–generalization argument.*

The BV condition typically holds when per-layer operator drifts are uniformly bounded (e.g., spectral-norm–regularized projections with stable LayerNorm scaling), yielding $\sum_{\ell=1}^{L} \|\Delta z^{(\ell)}\|_2 < \infty$; see App. Sections K and K.1 for worked examples, failure modes, and an online detection algorithm (Algorithm 1).

**Norm compatibility and error budget.** To interface with the mobility bounds in Section 2.2, we upper bound layer increments with $\|\cdot\|_\infty$ (comparable to $\|\cdot\|_2$ on compacts). Define

$$\Xi_L := \alpha_1 \max_\ell \left\|\Delta z^{(\ell)}\right\|_\infty + \alpha_2 \sum_\ell \left\|\Delta z^{(\ell)}\right\|_\infty^2, \tag{2}$$

where $\alpha_1, \alpha_2$ depend only on $L_b, M_b, \Lambda_J, L_J$ from Section 2.2. **Norm equivalence for the budget.** On compact domains and fixed dimension, $\|\cdot\|_2$ and $\|\cdot\|_\infty$ are equivalent up to constants. Thus the worst-case single-layer term and the cumulative squared-variation term in equation 2 are consistent with the $\|\cdot\|_2$-based BV assumption in Assumption 3.1; see Appendix D for the explicit constants used in the proof of Theorem 3.1.

**Theorem 3.1** (Finite-depth error to PF–ODE). *Under Assumption 3.1 and the regularity in Section 2.2, let $p(t)$ solve the probability-flow ODE on $[0,1]$ with $p(0) = \lim_{L \to \infty} p^{(0)}$. Then there exists $\Gamma = \Gamma(L_b, M_b, \Lambda_J, L_J)$ such that*

$$\sup_{t \in [0,1]} \left\| p^{(\lfloor tL \rfloor)} - p(t) \right\|_1 \leq \Xi_L + \left(e^\Gamma - 1\right) \left\| p^{(0)} - p(0) \right\|_1,$$

*with $\Xi_L$ in equation 2. In particular, if $p^{(0)} = p(0)$ and $\Xi_L \to 0$, then $p^{(\lfloor tL \rfloor)} \to p(t)$ uniformly in $t$.*

*Proof sketch.* The proof controls the per-layer error $\Delta_\ell = \|p^{(\ell)} - p(t_\ell)\|_1$ between the discrete stack and the PF–ODE at times $t_\ell = \ell/L$.

**Step 1: Local truncation.** A Taylor expansion of the PF–ODE solution around $t_\ell$, combined with stability estimates for the simplex Jacobian $J_{\mathrm{sm}}$ and drift $b$, yields a one-step inequality

$$\Delta_{\ell+1} \ \leq \ (1 + C_1 \delta t)\, \Delta_\ell + C_2 \|\Delta z^{(\ell)}\|_\infty \delta t + C_3 \delta t^2,$$

with $C_1 = L_b \Lambda_J + M_b L_J$ capturing Lipschitz and mobility bounds.

**Step 2: Global accumulation.** Iterating over all layers and applying a discrete Grönwall lemma (Lemma D.1) produces

$$\Delta_L \ \leq \ e^{C_1} \Delta_0 + e^{C_1} \big( C_2 C_{\mathrm{BV}} + C_3 L^{-1} \big),$$

where $C_{\mathrm{BV}} = \frac{1}{L} \sum_\ell \|\Delta z^{(\ell)}\|_\infty$ is the averaged bounded-variation constant.

**Step 3: Budget definition.** The finite-depth budget $\Xi_L$ in equation 2 combines worst-case jumps and cumulative squared variation, with explicit constants depending only on $(L_b, M_b, \Lambda_J, L_J)$. This yields the stated bound with $\Gamma = C_1$. $\square$

Complete details including explicit constant derivations and norm equivalence appear in Appendix D.

**Remark 3.2** (Continuum validity and constant scaling). *$\Xi_L$ is a practical validity threshold: the PF–ODE faithfully predicts layerwise behavior when $\Xi_L$ is small (see proof above; additional technical details in App. D). Moreover, the budget constants scale with architectural smoothness and geometry: $\alpha_1 = \mathcal{O}(L_b + M_b)$ and $\alpha_2 = \mathcal{O}(\Lambda_J + L_J)$. Hence $\Xi_L$ decreases with smaller per-layer logit increments and stronger contraction, and the PF–ODE discrepancy vanishes as $L \to \infty$ under fixed budgets.*

**Remark 3.3** (When BV holds in practice). *BV typically holds during stable training but can fail at phase transitions, early layers, or gradient instability. Detect via $S_L = \sum_\ell \|\Delta z^{(\ell)}\|_2^2$; if BV fails, apply piecewise analysis (App. D).*

**Lemma 3.4** (Compactness and absolute continuity). *Under Assumption 3.1, there exists a subsequence with $z_L \to z$ and $p_L \to p$ in $L^1([0,1])$ and a.e., where $p$ is absolutely continuous with $|\dot{p}| \in L^1$. The compactness and identification statements follow from Theorem J.1 in Appendix J.1.*

**Lemma 3.5** (Drift identification via architectural consistency). *Under Assumption 3.1 with (i) bounded total variation and uniform boundedness and (ii) architectural consistency, define $D_L(t) := \Delta z^{(\ell)} / \delta t$ on $[t_\ell, t_{\ell+1})$, and let $\hat{b}_L$ be the local regression estimator fit to the same layer transitions. Then*

$$\|D_L - \hat{b}_L\|_{L^1([0,1])} \to 0 \qquad and \qquad \|\hat{b}_L - b\|_{L^1([0,1])} \to 0,$$

*hence $\|D_L - b\|_{L^1([0,1])} \to 0$ and, in particular, $D_L \rightharpoonup b$ in $L^1([0,1])$. Proof sketch. **Calibration:** $D_L$ and $\hat{b}_L$ are computed from identical transitions, so regression residuals control $\|D_L - \hat{b}_L\|_{L^1}$. **Generalization:** architectural consistency yields $\|\hat{b}_L - b\|_{L^1} \to 0$ on compacts. Triangle inequality concludes. (Complete details appear in Appendix J.1.)*

### 3.2 Semi-relaxed optimal transport and causal attention

**Remark 3.6** (Row-softmax via semi-relaxed EOT). *By the KL-prox characterization in Proposition 2.1, standard row-softmax solves a semi-relaxed entropic OT step (with masking handled by infinite costs and restricted support). We refer to Appendix B for details of the dual and masking.*

### 3.3 Probability-flow ODE emergence and well-posedness

**Theorem 3.7** (PF–ODE on the simplex and well-posedness). *Under Assumption 3.1 and the regularity in Section 2.2, the limit probability path satisfies*

$$\dot{p}(t) = J_{\mathrm{sm}}(z(t))\, b(z(t), t) \quad a.e. \ on \ [0,1], \qquad p(0) = \lim_{L \to \infty} p^{(0)},$$

and the velocity field $v(p, t) = J_{\mathrm{sm}}(z(t)) \, b(z(t), t)$ is tangent to the simplex, ensuring $p(t) \in \Delta^{V-1}$ for all $t$.

*Complete proof in Appendix Q.1.*

**Remark 3.8** (Simplex invariance and uniqueness). *Under Carathéodory conditions on $b$ (measurable in $t$, locally Lipschitz in $z$), mass is conserved ($\sum_i p_i(t) = 1$), nonnegativity holds, zero-flux $J_{\mathrm{sm}}(z)\mathbf{1} = 0$ enforces boundary behavior, and solutions are unique on the relative interior of $\Delta^{V-1}$.*

**Theorem 3.9** (Locking via vanishing mobility). *If $p_{\max}(t) \to 1$ and $b$ is bounded, then $\|J_{\mathrm{sm}}(z(t))\|_{\mathrm{op}} \to 0$ (Remark 2.2) and hence $\|\dot{p}(t)\| \to 0$. Moreover, temperature rescales mobility as $J_{\mathrm{sm}}^{(\tau)}(z) = \frac{1}{\tau} J_{\mathrm{sm}}(z/\tau)$, modulating the approach to locking.*

**Example 3.10** (Two-token mobility collapse (summary)). *Consider a minimal attention layer with two tokens. For logits $z = (z_1, z_2)$, the softmax Jacobian reduces to*

$$J_{\mathrm{sm}}(z) = p(1-p) \begin{pmatrix} 1 & -1 \\ -1 & 1 \end{pmatrix},$$

*where $p = p_1 = \mathrm{softmax}(z)_1$. The operator norm is $\|J_{\mathrm{sm}}(z)\|_{\mathrm{op}} = 2\, p(1-p)$, which attains its maximum $1/2$ at the uniform distribution $p = 1/2$ and collapses to zero as $p \to 0$ or $p \to 1$. With temperature $\tau$, the effective mobility scales as $\tau^{-1} p(1-p)$.*

*As attention mass locks onto one token, the mobility eigenvalue vanishes, forcing $\dot{p} = J_{\mathrm{sm}}(z)\, b(z, t)$ to approach zero even if the drift $b$ remains nonzero. Temperature rescaling modulates this: larger $\tau$ maintains nontrivial mobility deeper into the network. This illustrates the mechanism behind Theorem 3.9: as distributions concentrate, the mobility tensor loses rank and dynamics freeze for geometric reasons, not because the drift disappears. Complete eigenvalue calculations appear in Appendix L.3.*

### 3.4 Connections to empirically observed phenomena

Attention entropy collapse, temperature scaling effects, and representation collapse follow naturally from the mobility interpretation: as distributions concentrate, mobility (and thus velocity) vanishes (Theorem 3.9), explaining attention concentration and providing a handle for calibration via temperature scaling. We defer expanded discussion, diagnostics, and eigenspectrum-based tests to Appendix E.

## 4 Diffusion Duality with Anisotropic Noise

### 4.1 Stochastic dynamics and weak Fokker–Planck formulation

We extend the probability-flow picture to include stochastic perturbations, establishing a duality between deterministic and stochastic evolution. Consider the hidden-state SDE:

$$dH_t = F(H_t, t) \, dt + \Sigma(H_t, t) \, dW_t, \tag{3}$$

with diffusion tensor $a = \Sigma\Sigma^\top$. Our analysis accommodates minimal regularity ($F$ locally integrable with weak derivatives, $a$ measurable and locally bounded), anisotropy ($a$ may be degenerate or near-singular), and time-inhomogeneity.

**Lemma 4.1** (Distributional calculus in weak FP regime). *Under local Fisher-information conditions ($p_H > 0$ a.e., $p_H \nabla \log p_H \in L^1_{\mathrm{loc}}$), the product rule holds distributionally:*

$$\nabla \cdot \nabla \cdot (a\, p_H) = \nabla \cdot \big((\nabla \cdot a)\, p_H + a\, \nabla p_H\big) \quad \text{in } \mathcal{D}'.$$

*Proof via mollification and weak convergence in Appendix F.*

**Theorem 4.2** (PF–ODE / reverse-SDE duality). *Let $a(x, t) = \sigma(x, t)\sigma(x, t)^\top$ and suppose $p_H(\cdot, t) > 0$ solves the Fokker–Planck equation*

$$\partial_t p_H \;=\; -\nabla \cdot (F\, p_H) \;+\; \tfrac{1}{2} \sum_{i,j} \partial_{x_i x_j}\big(a_{ij}\, p_H\big)$$

*with suitable decay/no-flux boundary conditions. Define the deterministic flow*

$$u(x,t) \;=\; F(x,t) \;-\; \tfrac{1}{2}\Big(a(x,t)\,\nabla_x \log p_H(x,t) \;+\; (\nabla\cdot a)(x,t)\Big), \qquad (4)$$

*where $(\nabla\cdot a)_i := \sum_j \partial_{x_j} a_{ij}$. Then the PF–ODE with velocity $u$ shares identical marginals with the Itô SDE for all $t$. If $a \equiv 2\beta I$ is spatially constant, then $u = F - \beta\,\nabla \log p_H$ recovers the standard probability flow drift. Complete proof in Appendix F.*

**Corollary 4.3** (Simplex marginal preservation). *For the softmax projection $\varphi(h) = \mathrm{softmax}(W^\top h)$, the pushforward measures satisfy $\varphi_{\#} p_H(\cdot,t) = \varphi_{\#}\rho(\cdot,t)$ a.e. in time, extending the duality to simplex-valued processes. Proof in Appendix F.*

**Proposition 4.4** (Anisotropy propagation to simplex dynamics). *The hidden-space diffusion induces an effective mobility on the simplex: $M(p) = J_{\mathrm{sm}}(z)\,W^\top a\,W\,J_{\mathrm{sm}}(z)$, revealing how architectural choices modulate probability dynamics. Proof in Appendix F.*

### 4.2 Weak approximation of diffusion by stacked attention

**Theorem 4.5** (Weak SDE approximation by stacked attention). *Under the assumptions in Section 2.2 and the weak FP calculus of Lemma 4.1, let $\rho(t)$ be the law of the reverse SDE with drift $u$ in equation 4 and diffusion $a$, and let $\widehat{\rho}_L(t)$ be the law induced by $L$ stacked attention layers with step $\delta t = 1/L$. Then, for any $\phi \in C_b^2$ and $T \in [0,1]$,*

$$\Big|\mathbb{E}_{\widehat{\rho}_L(T)}[\phi] - \mathbb{E}_{\rho(T)}[\phi]\Big| \;\leq\; C_\phi\Big(L^{-1} + \max_{0\leq \ell < L}\|\Delta z^{(\ell)}\|_\infty + \gamma\Big),$$

*where $C_\phi$ depends on bounds of $u, a$ and $\phi$ on compacts, and $\gamma \geq 0$ is an optional degeneracy regularizer. Proof in Appendix F.*

Stacked attention approximates anisotropic, time-inhomogeneous diffusion in a weak sense; the approximation error separates discretization, logit variation, and degeneracy regularization. Anisotropic diffusion with widely varying eigenvalues induces directional stiffness mirroring attention's collapsed-coordinate behavior, explaining curvature and locking diagnostics. Toy example and degeneracy guidelines in Appendix F.

## 5 Schrödinger Bridges and Transport Optimality

### 5.1 General framework and alignment conditions

Schrödinger Bridges (SB) characterize entropy-regularized stochastic interpolations between endpoint distributions. We establish when transformer-induced probability paths align with these optimal bridges. While Section 4 allows degenerate diffusion (useful near locking), SB typically requires a uniformly elliptic reference; we reconcile these views below.

**Assumption 5.1** (Reference diffusion). *The reference process $R$ follows $dX_t = b_R(X_t,t)\,dt + \sigma(X_t,t)\,dW_t$ with diffusion tensor $a = \sigma\sigma^\top$, where:*

1. *Non-degeneracy on support: $a(x,t)$ is SPD almost everywhere on the support of the path measure.*
2. *Finite action: The reference path has finite relative entropy with respect to Wiener measure for endpoints $(\mu_0, \mu_1)$.*
3. *Degeneracy handling (regularization): When $a$ approaches singularity (e.g., near locking), we use $a_\varepsilon = a + \varepsilon I$, analyze with $\varepsilon > 0$, and pass to the limit $\varepsilon \downarrow 0$ (see Appendix G).*

**Theorem 5.1** (SB alignment characterization). *Let $\{\mu_t\}_{t\in[0,1]}$ be the transformer's continuous-depth probability path with drift $u$. Under Assumption 5.1, $\{\mu_t\}$ equals the Schrödinger Bridge for reference $R$ if and only if its per-mass velocity decomposes as*

$$u \;=\; b_R \;+\; a\,\nabla\theta$$

*for some potential $\theta$. Equivalently, the $a$-weighted curl vanishes, i.e. the solenoidal component of $a^{-1}(u - b_R)$ is zero. A proof is provided in Appendix G.*

**Theorem 5.2** (Rotational energy controls SB deviation). *Let $u = b_R + a\nabla\theta + w$ be the $a$-weighted Hodge decomposition with $\nabla \cdot (w\,\mu_t) = 0$ for each $t$. Define the rotational energy*

$$\mathcal{R} = \int_0^1 \int \langle w, a^{-1}w \rangle \, \mu_t(\mathrm{d}x) \, \mathrm{d}t.$$

*Assume a finite weighted Poincaré constant $C_P(\mu, a)$ along the path. Then, for each $t \in [0, 1]$,*

$$\mathrm{KL}\big(\mu_t \,\|\, \mu_t^\star\big) \leq C_P(\mu, a)\,\mathcal{R},$$

*where $\mu_t^\star$ is the SB path with the same endpoints and reference $R$. In particular, $\mathcal{R} = 0$ if and only if $\{\mu_t\}$ is SB-aligned. A proof is given in Appendix G.*

**Corollary 5.3** (Rotational energy diagnostic). *$\mathcal{R} \geq 0$ with equality iff the path is Schrödinger Bridge. Practically, estimate $u$ (from activations), solve for $\theta$ via a weighted Poisson equation, compute the residual $r = u - b_R - a\nabla\theta$, and evaluate $\int \|a^{-1/2}r\|^2 \, \mathrm{d}\mu \, \mathrm{d}t$. See App. Figure 6 for a compact schematic of this pipeline.*

**Example 5.4** (Rotational energy on the two-simplex). *Consider a toy flow on $\Delta^2 = \{(p_1, p_2, p_3) : p_i \geq 0, \sum_i p_i = 1\}$ with coordinates $(p_1, p_2)$ and $p_3 = 1 - p_1 - p_2$. Let $a = \sigma^2 I$ and $b_R = 0$. A gradient flow $u = a\nabla\theta$ with $\theta(p_1, p_2) = -\frac{\alpha}{2}(p_1^2 + p_2^2)$ yields $u = (-\alpha\sigma^2 p_1, -\alpha\sigma^2 p_2)$. The curl vanishes:*

$$\partial_{p_1} u_2 - \partial_{p_2} u_1 = 0,$$

*confirming zero rotational energy and pure Schrödinger Bridge structure. Conversely, the rotational flow $u = (\beta p_2, -\beta p_1)$ circulating around the simplex center yields*

$$\partial_{p_1}(-\beta p_1) - \partial_{p_2}(\beta p_2) = -2\beta \neq 0,$$

*indicating rotational energy proportional to $\beta^2$. This flow cannot arise as a Schrödinger Bridge because it lacks potential structure; the rotational component represents spurious circulation that wastes transport capacity on cycles rather than moving mass toward the terminal distribution. Our diagnostic computes an empirical analogue by discretizing the flow field from layer activations, approximating spatial derivatives via finite differences, and integrating across the trajectory. Small $\mathcal{R}$ indicates approximate gradient structure and Schrödinger Bridge alignment; large $\mathcal{R}$ reveals spurious rotational components deviating from optimal transport.*

**Remark 5.5** (Vanishing-regularization limit). *If $a_\varepsilon \to a$ with $\varepsilon \downarrow 0$ and the sequence of SB paths has uniformly bounded action and is tight, any weak limit is a degenerate SB solution; when $\mathcal{R} = 0$, it coincides with the PF–ODE path. See Appendix G.*

**Corollary 5.6** (Simplex Schrödinger Bridge). *Under the softmax pushforward, the SB condition on the simplex takes the potential-flow form*

$$\dot{P}_t = -\nabla_p \cdot \big(P_t \, M(P_t) \, \nabla_p \Theta(P_t, t)\big),$$

*with mobility $M$ from Theorem 4.4. This connects directly to gradient flows on the simplex and informs mobility-aware design.*

**Practical implication.** Rising $\mathcal{R}$ indicates deviation from SB (OT) geometry and co-occurs with over-smoothing and spurious drift; minimizing $\mathcal{R}$ provides a geometry-aware early warning complementary to standard fidelity metrics.

## 6 EMPIRICAL VALIDATION FRAMEWORK AND DIAGNOSTIC TOOLS

**Overview and theory map.** We validate two tracks: *(T)* Transformers (forward pass as PF–ODE) and *(I)* image diffusion (parity and weak-error).

**Probability-flow ODE dual (summary).** In variance-preserving (VP) score-based diffusion, the forward SDE is $dx = f(t)\,x\,dt + g(t)\,dW_t$ and the learned score $\nabla_x \log p_t(x)$ defines a deterministic probability-flow ODE (PF–ODE) with drift $f(t)\,x - \frac{1}{2}g(t)^2\nabla_x \log p_t(x)$ that shares the SDE time marginals.

We use the formal definitions from App. Section M.1 for the drift budget, locking bound, curvature, and EVI (Equations (22) to (25)) throughout this section. Drift and curvature visualizations appear in Figure 1 (left/right panels), while locking and EVI are shown in App. Figures 7 and 8.

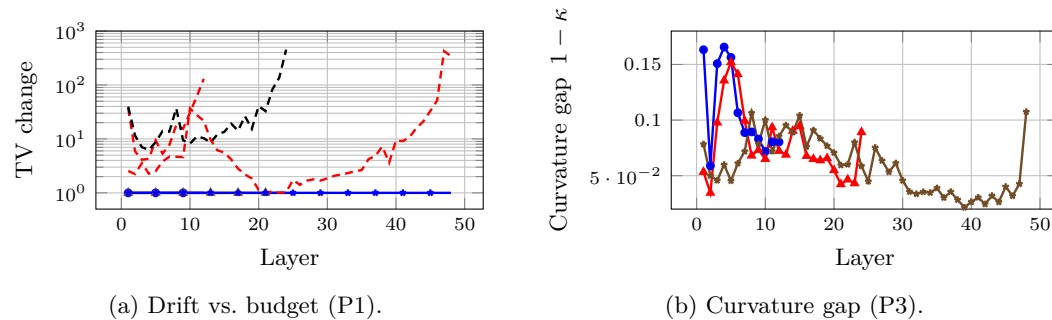

(a) Drift vs. budget (P1).

(b) Curvature gap (P3).

Figure 1: Track T: core diagnostics. Left: PF–ODE adequacy (P1). Right: curvature (P3). Locking and EVI appear in Section M.4.

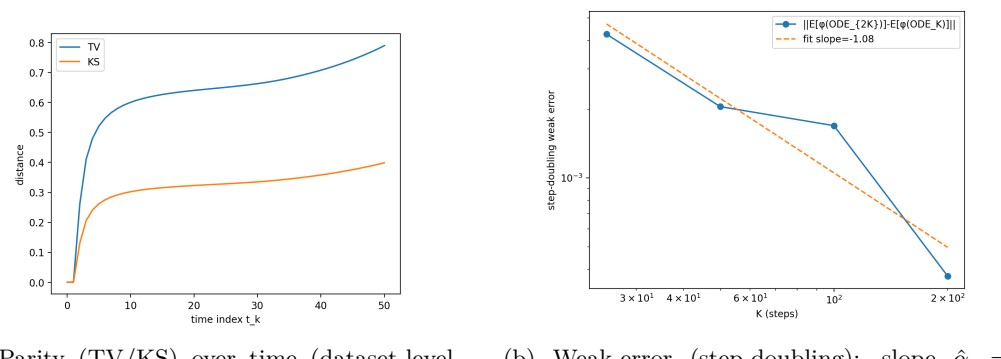

(a) Parity (TV/KS) over time (dataset-level, per-channel histograms).

(b) Weak-error (step-doubling); slope $\hat{\alpha} = -1.08$ (95% CI $[-2.18, -0.28]$).

Figure 2: Image diffusion (CIFAR-10). Left: ODE–SDE TV/KS across time (inputs scaled to $[0,1]$, equal channel weighting, 256 bins). Right: log–log regression of $\Delta_K$ vs. $K$ with BCa CIs (B=1000).

## 6.1 Empirical diagnostics P0–P4

**Diagnostics (P0–P4).** Five diagnostics validate the theory: (P0) BV/continuity checks; (P1) PF–ODE adequacy; (P2) locking behavior; (P3) OT contractivity; (P4) SB alignment via rotational energy. Full protocols appear in App. Section M.

## 6.2 Track T: Transformers – core diagnostics and rotational energy

For the Transformer experiments, the mean rotational energy across 10 central layers is $\widehat{\mathcal{R}} = 1.096 \times 10^{-7}$ (95% CI $[3.468 \times 10^{-8}, 2.153 \times 10^{-7}]$). Cross-track values are not comparable due to different ambient spaces and discretizations; we summarize per-track means and CIs (a normalized variant is defined in the appendix).

## 6.3 Track I: Image diffusion—parity, weak-error, and SB energy

**Setup.** A trained VP CIFAR-10 (ddpm++ continuous) model is evaluated with two samplers: SDE and PF–ODE; both samplers use the identical noise schedule and classifier-free guidance setting, and for each image the ODE and SDE share the same initial noise seed. We use $N$=10,000 images and $K$=50 logged times on a shared grid.

**Parity and weak-error (composite).** Figure 2 composes the image diagnostics: left shows ODE–SDE histogram parity (TV/KS) over time; right shows the weak-error step-doubling log–log fit (slope near first order).

**Rotational energy (image; P4 result).** On 20 time points, the mean rotational energy is $\widehat{\mathcal{R}} = 0.03092$ (95% CI $[0.01046, 0.05385]$). Cross-track values are not comparable due to different ambient spaces and discretizations; per-track normalized variants and the BV panel for ODE vs. SDE appear in App. Section M.5.

**Defaults.** Unless noted, for the image track PF–ODE uses deterministic sampling on the same $K$ grid as SDE (DDIM-style); for Transformers, PF–ODE drift fits use Dormand–Prince with $\texttt{rtol}= 10^{-5}$, $\texttt{atol}= 10^{-7}$, $\texttt{max\_steps}= 2000$. Ridge grid $\{10^{-4}, 10^{-3}, 10^{-2}\}$ with 5-fold cross-validation; $N = 50\text{k}$ rows/layer (Transformers), $N = 10\text{k}$ images and $K = 50$ time steps (Image). Unless noted, bands denote 95% *percentile*-bootstrap CIs (B=200); weak-error CIs use BCa (B=1000).

**Synthesis.** Taken together, the two empirical tracks support a single underlying picture: attention dynamics in Transformers and PF–ODE/SDE trajectories in diffusion models behave as different discretizations of the same entropy-regularized transport flow. Locking and EVI signatures appear in the appendix; the core P1/P3 diagnostics remain in the main text.

## 7 Limitations and Practical Implications

**Limitations.** (i) Experiments target *text* transformers with a minimal image diffusion sanity check; full vision benchmarks are out of scope for this paper (Section N.1). (ii) The PF–ODE drift uses simple features and can underfit nonlocal effects. (iii) Rotational-energy magnitudes are track-specific and not cross-track comparable; we provide a dimensionless variant for intra-track comparison and recommend log-scale plots when ranges span orders of magnitude (App. Section N.2). (iv) Diagnostics are conditioned on the P0 gate (BV/-continuity); failures trigger abstention.

**Practical implications and Outlook** (1) Temperature or key-norm controls reduce the curvature gap $1 - \kappa$, offering a stable knob for depth behavior. (2) The drift-budget overlay surfaces over-activation and can inform regularization or early exit policies. (3) Diagnostic abstention protocols based on P0 gating conditions provide conservative guardrails when validity assumptions are violated (Section N.4). Richer drift features (e.g., cross-head structure), broader modalities beyond CIFAR-10, structured/accelerated SB solvers, and calibration via condition-number targets for the Poisson step are natural directions (Section O).

## 8 Conclusion

We formalized masked attention as semi-relaxed entropic OT, established stability/locking and curvature/EVI structure with gauge invariances, and tied these to a practical empirical suite. The suite validates PF–ODE adequacy, locking signatures, and contractivity response in Transformers, and shows image PF–ODE/SDE parity with first-order weak-error scaling. These yield concrete levers (temperature/key norm; drift-informed regularization) and geometry-aware diagnostics; extended discussion appears in Section O. For practitioners: (i) regulate depth via $\Xi_L$/stability budgets and spectral norm controls, (ii) monitor rotational energy during schedule sweeps as an early-warning diagnostic, and (iii) abstain when P0 validity conditions fail.

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

**Ethics Statement.** We have read and will adhere to the ICLR Code of Ethics. This work develops a theoretical and diagnostic framework unifying transformers and diffusion models via entropy-regularized optimal transport (OT). Our experiments use only publicly available datasets and open checkpoints where licenses allow redistribution or scripted download (details in the appendix). We do not collect, annotate, or release any personal or sensitive data, and we do not deploy models for user-facing decisions. The proposed diagnostics (P0–P4), PF–ODE integration, and entropy-based temperature scheduling are intended to improve scientific understanding and training/serve-time efficiency (e.g., early exit). Potential risks are limited to misinterpretation or over-generalization of the diagnostics outside their validity regime (e.g., when bounded variation fails or under heavy sparsity/MoE routing). To mitigate this, we clearly document assumptions, abstain when diagnostic preconditions fail, and report limitations (mixture-of-experts, highly sparse attention, and early training phases). We see no domain-specific legal, privacy, or safety issues introduced by this study.

**Reproducibility Statement.** We aim for complete reproducibility. The appendix specifies: (i) data sources, splits, and licenses; (ii) model checkpoints and versions; (iii) all hyperparameters; (iv) exact diagnostic protocols; (v) hardware and runtime details. Upon acceptance, we will release a repository containing:

- **Diagnostics (P0–P4).** Implementations for BV/continuity checks (P0), PF–ODE adequacy and drift fitting (P1), locking and curvature/EVI (P2–P3), and rotational energy / SB diagnostic (P4), including numerically stable Poisson solves and $a$-weighted Hodge decomposition.

- **PF–ODE Integration.** Reference ODE solvers with error control and scripts to compare ODE vs. SDE marginals for the duality experiments.

- **Weak-Error Evaluation.** Step-doubling protocol with BCa bootstrap ($B$=1000) and log–log slope estimation; code to reproduce the reported confidence intervals.

- **Image Parity (Track I).** TV/KS histogram parity evaluation on CIFAR-10 with $N$=10,000 images and $K$=50 time points, including seeds and preprocessing.

- **Entropy-Based Temperature Scheduling.** Continuous and discrete schedules (EMA, clipping bounds) with ablation hooks.

- **Configuration + Seeds.** YAML configs for each experiment, fixed random seeds, and deterministic flags where supported by the backend.

We provide scripts to fetch datasets and (where licenses permit) checkpoints, plus a manifest of software versions (CUDA/driver, PyTorch/JAX, Python), GPU type, and expected wall-clock ranges. Plots are generated from saved CSV logs to ensure exact figure reproduction. The repository will include a one-command orchestration to reproduce paper artifacts end-to-end.

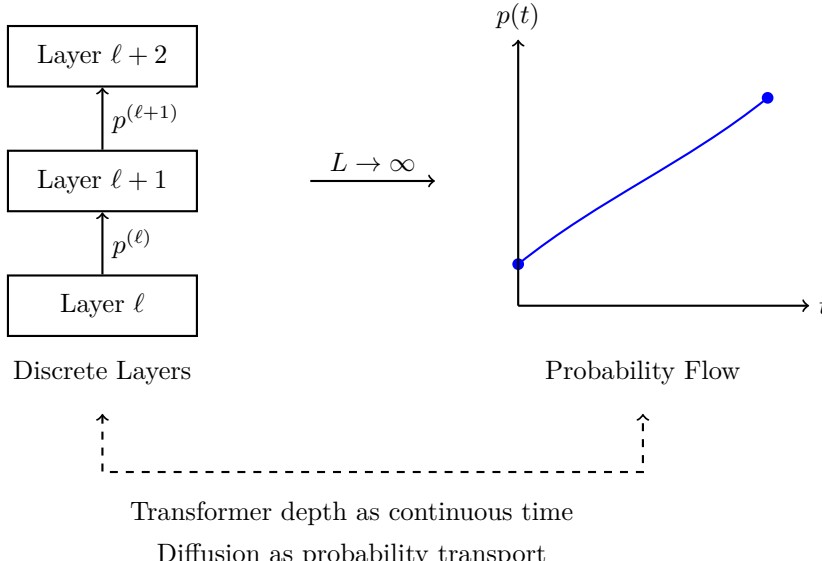

Figure 3: Conceptual unification: transformer layers implement discrete steps of probability transport that converge to continuous flows analogous to diffusion models. The softmax normalization induces entropic regularization, while layer stacking corresponds to time evolution.

## A    Supplementary Motivation and Overview

**Extended motivation.**    The remarkable success of transformers in language modeling and diffusion models in generation has driven rapid progress in artificial intelligence, yet our theoretical understanding of these architectures remains fragmented. Transformers process discrete tokens through attention mechanisms that mysteriously develop semantic understanding, while diffusion models generate high-quality samples through iterative refinement processes that seem fundamentally different. This theoretical gap impedes principled architectural improvements and forces practitioners to rely on empirical trial-and-error rather than systematic design principles. In this work, we demonstrate that these seemingly disparate architectures are actually implementing the same fundamental computational principle: entropy-regularized optimal transport of probability mass. This unification not only explains numerous empirical phenomena that have puzzled researchers but also provides concrete tools for improving both architectures.

Modern deep learning relies heavily on two architectural paradigms: transformers, which dominate language modeling through attention-based token mixing, and diffusion models, which excel at generation through iterative denoising. Despite their apparent differences— transformers operate on discrete tokens with normalized attention weights, while diffusion models evolve continuous densities through stochastic differential equations—we demonstrate that both architectures implement entropy-regularized transport of probability mass.

**Interpretive notes.**    The significance of this connection extends beyond theoretical curiosity. Understanding transformers and diffusion models as implementing the same fundamental transport process enables principled architectural improvements and explains puzzling empirical phenomena. For instance, the widespread observation that attention patterns become increasingly concentrated in deeper transformer layers, often leading to computational waste, can now be understood as a geometric inevitability arising from the vanishing mobility of the softmax-induced transport. Similarly, the empirical success of temperature scaling for improving model calibration emerges naturally from our framework as a mobility modulation mechanism. By revealing these deep structural connections, our framework provides actionable insights for model design: predicting when representations will lock, identifying optimal depth for different tasks, and suggesting principled initialization strategies that approximate continuous optimal transport paths.

| Framework Overview: From Theory to Practice | |
|---|---|
| *Theoretical Concept* | *Practical Implication* |
| Softmax Jacobian as mobility tensor $J_{\mathrm{sm}}$ | Quantifies capacity for probability updates; vanishing mobility signals when to stop computation |
| Bounded variation regime $S_L < C$ | Smooth evolution enables continuous analysis; violations indicate phase transitions requiring intervention |
| Semi-relaxed EOT preserves causality | Maintains autoregressive structure while enabling optimal transport analysis of attention |
| Probability-flow ODE limit | Suggests continuous-depth architectures and adaptive depth selection based on task complexity |
| Schrödinger Bridge alignment | Rotational energy $\mathcal{R}$ measures deviation from optimality, guiding architectural improvements |
| Anisotropic diffusion duality | Reveals how noise injection affects transport; suggests principled dropout and regularization strategies |

*Key Diagnostics:*

- *During Training:* Monitor $S_L$ for stability, $\|J_{\mathrm{sm}}\|$ for representation health

- *Architecture Design:* Use $\mathcal{R}$ to compare transport efficiency across architectures

- *Deployment:* Apply mobility thresholds for early exit decisions

Figure 4: Overview linking theory to practice. Each theoretical concept maps to a concrete tool or diagnostic.

| | Balanced OT (Sinkhorn) | Semi-relaxed OT (ours) | Diffusion / SB |
|---|---|---|---|
| Causality preserved | No | Yes | Yes |
| Depth → continuum | Heat flow | PF–ODE on simplex | FP / PF–ODE |
| Noise model | — | Anisotropic via FP | General $a$ (SB) |
| SB equivalence (iff) | No | Yes | Yes |
| Locking mechanism | — | $J_{\mathrm{sm}} \to 0$ | Entropy collapse |

Table 1: Novelty map relative to prior strands. Semi-relaxed EOT preserves the causal structure essential for autoregressive models while enabling rigorous continuous-depth analysis. The vanishing of $J_{\mathrm{sm}}$ provides a geometric explanation for attention collapse.

# B   SUPPLEMENTARY PROOFS AND TECHNICAL DETAILS

**Proof of the sharp mobility bound (Remark 2.2).** Let $p = \mathrm{softmax}(z/\tau)$ and $J_{\mathrm{sm}}(z) = \mathrm{Diag}(p) - pp^\top$. Then $J_{\mathrm{sm}}$ is symmetric and positive semidefinite on the simplex tangent space. For any unit vector $v$ with $\sum_i v_i = 0$,

$$v^\top J_{\mathrm{sm}} v \;=\; \sum_i p_i v_i^2 \;-\; \left( \sum_i p_i v_i \right)^2 \;\le\; \tfrac{1}{2} \sum_i p_i v_i^2,$$

with equality achieved for distributions supported on two atoms at mass $\frac{1}{2}$ and $v$ aligned with that two-dimensional subspace. Scaling $z \mapsto z/\tau$ yields the factor $1/\tau$, hence $\|J_{\mathrm{sm}}(z)\|_{\mathrm{op}} \le \frac{1}{2\tau}$ and the spectrum is contained in $[0, \frac{1}{2\tau}]$, collapsing to $\{0\}$ as $p_{\max} \to 1$. $\qquad\square$

**Semi-relaxed EOT details.** We provide the complete derivation of the semi-relaxed entropic optimal transport characterization of attention.

Let $q \in \mathbb{R}^{d_k}$ be the query vector and $k_j \in \mathbb{R}^{d_k}$ the $j$-th key vector in a shared key-query space. Define the cost $c_j = -q \cdot k_j$, so that high similarity corresponds to low transport cost. Given a reference distribution $u \in \Delta^{V-1}$ (typically uniform), we consider

$$\min_{p \in \Delta^{V-1}} \left\{ \sum_{j=1}^{V} p_j c_j + \tau \, \mathrm{KL}(p\|u) \right\}, \tag{2.1}$$

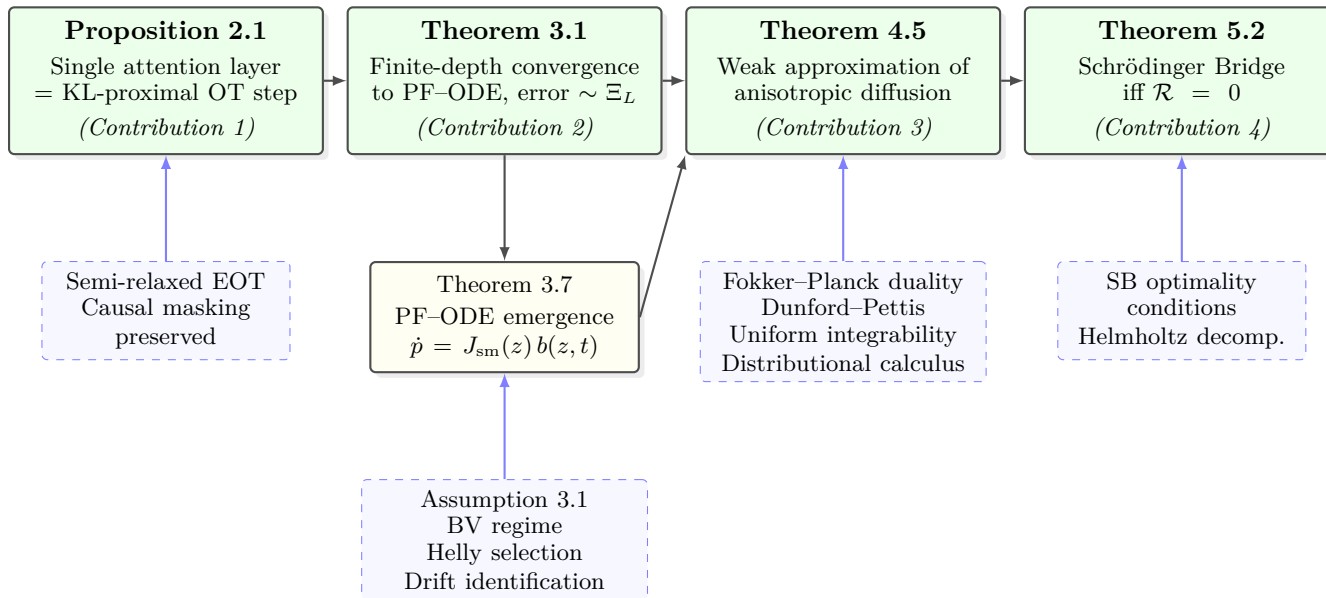

Figure 5: Logical structure and dependencies of main theoretical results. The framework establishes four contributions enumerated in Section 1: **(1)** Proposition 2.1 shows that a single attention layer implements a KL-proximal optimal transport step in the sense of Jordan-Kinderlehrer-Otto schemes, establishing the foundational connection between neural architecture and optimal transport geometry. **(2)** Theorem 3.1 proves that stacked attention layers converge to continuous probability flows on the simplex with explicit finite-depth error bounds controlled by the bounded variation budget $\Xi_L$, using Helly selection and architectural consistency to identify the limiting drift. **(3)** Theorem 4.5 demonstrates that the limiting probability flow weakly approximates time-inhomogeneous anisotropic reverse diffusions, unifying attention and diffusion through Fokker-Planck duality. **(4)** Theorem 5.2 provides a rotational energy characterization showing that vanishing $\mathcal{R}$ is necessary and sufficient for exact Schrödinger Bridge alignment. The intermediate result Theorem 3.7 establishes well-posedness of the probability flow ODE with the softmax Jacobian as mobility tensor. Blue dashed boxes indicate foundational assumptions and mathematical tools; solid boxes with shadows indicate proven results. Green highlighting emphasizes the four main contributions; yellow indicates supporting infrastructure.

with temperature $\tau > 0$ and $\mathrm{KL}(p\|u) = \sum_j p_j \log(p_j/u_j)$.

**Step 1: Lagrangian formulation.** Imposing $\sum_j p_j = 1$ via Lagrange multiplier $\lambda$ gives

$$\mathcal{L}(p,\lambda) = \sum_{j=1}^{V} p_j c_j + \tau \sum_{j=1}^{V} p_j \log \frac{p_j}{u_j} - \lambda\Big(\sum_{j=1}^{V} p_j - 1\Big).$$

**Step 2: First-order optimality conditions.** Setting $\partial \mathcal{L}/\partial p_j = 0$ yields

$$c_j + \tau\Big(\log \frac{p_j}{u_j} + 1\Big) - \lambda = 0.$$

Solving for $p_j$ gives

$$p_j = u_j \exp\Big(\frac{\lambda - c_j - \tau}{\tau}\Big) = C\, u_j \exp\Big(-\frac{c_j}{\tau}\Big),$$

where $C = \exp((\lambda - \tau)/\tau)$ is a normalization constant.

**Step 3: Row normalization yields softmax.** Enforcing $\sum_j p_j = 1$ determines $C$:

$$C = \Big[\sum_{j=1}^{V} u_j \exp(-c_j/\tau)\Big]^{-1}.$$

For uniform $u_j = 1/V$, we obtain

$$p_j = \frac{\exp(-c_j/\tau)}{\sum_{k=1}^{V} \exp(-c_k/\tau)} = \frac{\exp(q \cdot k_j/\tau)}{\sum_{k=1}^{V} \exp(q \cdot k_k/\tau)},$$

which is exactly the standard attention weight $p_j = \text{softmax}(qK^\top/\tau)_j$.

Thus each attention row solves a semi-relaxed entropic OT problem where the row-stochastic constraint is enforced but column marginals are unconstrained, preserving autoregressive structure. Causal masking is implemented by assigning infinite cost $c_j = +\infty$ to masked positions; the resulting row-normalized solution coincides with the attention distribution induced by logits, with existence and uniqueness guaranteed by the strict convexity of the KL divergence.

**Proof of Proposition 2.1.** The mirror-descent Euler step in KL geometry with objective $\langle c, p \rangle$ and step $\tau$ yields the variational form in Proposition 2.1. The unique minimizer has Gibbs form relative to $u$, $p^+ \propto u \odot \exp(-c/\tau)$, matching attention with logits $z = -c$. Stacking steps gives a discrete JKO/Mirror scheme. □

## C EXTENDED RELATED WORK AND POSITIONING (FULL VERSION)

### C.1 PROBABILITY FLOWS AND SCHRÖDINGER BRIDGES

Score-based diffusion established that reverse-time SDEs admit a *probability–flow* ODE with identical marginals (Song et al., 2021), while flow matching proposed simulation-free training of vector fields that realize desired probability paths (including OT geodesics) (Lipman et al., 2022). The Schrödinger Bridge (SB) program casts diffusion as *entropic* OT on path space and provides scalable IPF-style solvers (De Bortoli et al., 2021; Shi et al., 2023). We leverage this geometry *inside* transformers: depth induces a PF–ODE on the simplex, weak/anisotropic FP theory gives a deterministic/stochastic duality for hidden-state evolution, and an *if&only-if* potential-plus-reference drift condition characterizes when a transformer's probability path is exactly an SB.

### C.2 ATTENTION AS ENTROPIC OPTIMAL TRANSPORT

Balanced OT views of attention enforce doubly-stochastic constraints via Sinkhorn iterations (Sander et al., 2022; Tay et al., 2020), and OT-based co-attention improves multimodal learning (Xu et al., 2023). A complementary line shows transformers can be *programmed to solve* entropic OT with accuracy improving in depth (Daneshmand, 2024). In contrast, we work in the *causal* regime and prove that *standard row-softmax attention* is precisely the optimizer of a *semi-relaxed* entropic OT (row constraints only), which preserves autoregressive masking and does not require imposing OT constraints at training time. From this equality we derive a BV depth→PF–ODE limit and the SB characterization in the causal setting; balanced OT results do not cover this regime and are fundamentally incompatible with the autoregressive structure essential to language modeling.

### C.3 CONTINUOUS-TIME VIEWS OF TRANSFORMERS

Continuous-depth interpretations of transformers address irregular time environments and ODE couplings (Zhang et al., 2021; Chen et al., 2023); OT-Transformer introduces OT as a *regularizer* in a continuous-time backbone (Kan et al., 2025). These works, however, do not *explicitly* endow the dynamics with an entropic-OT geometry that explains empirical phenomena. Our framework fills this gap: the softmax Jacobian acts as a *mobility tensor* on $\Delta^{V-1}$, depth induces a PF–ODE with simplex invariance and well-posedness, and SB equivalence provides a variational certificate for transport optimality.

### C.4 AUTOREGRESSIVE–DIFFUSION HYBRIDS

Bridging autoregressive and diffusion/flow paradigms has shown strong empirical results (Hoogeboom et al., 2022; Ma et al., 2025). Our theory explains *why*: AR transformers and

diffusion models are two discretizations (discrete in depth vs. continuous in time) of the same entropy-regularized transport principle. The PF–ODE/FP duality and SB tools provide quantitative diagnostics (e.g., rotational energy) for assessing alignment with entropic OT.

### C.5 ARCHITECTURAL UNIFICATION VIA DIFFUSION TRANSFORMERS

Replacing U-Nets with transformer backbones yields scalable diffusion models across images and 3D (Peebles & Xie, 2023; Mo et al., 2023). While these works focus on performance, our analysis rationalizes their success: both families implement transport under entropic regularization, and temperature/mobility schedules, anisotropy-aware regularization, and SB-aligned depth emerge as principled design levers independent of the backbone.

### C.6 POSITIONING OF OUR CONTRIBUTIONS

**(i) Causal, semi-relaxed OT for attention.** We prove that *unmodified* row-softmax attention solves a row-constrained entropic OT problem, resolving the incompatibility of balanced OT with causal masking.

**(ii) Depth → PF–ODE on the simplex.** Under bounded-variation scaling, stacking attention layers induces a PF–ODE for probe-induced probabilities, with simplex invariance and well-posedness.

**(iii) Weak FP duality with anisotropy.** Allowing time-inhomogeneous, anisotropic (and possibly ill-conditioned) diffusion, we establish deterministic/stochastic equivalence of marginals via Fokker–Planck in the renormalized/weak sense.

**(iv) SB equivalence (iff) & diagnostics.** The depth path is an SB iff its velocity is potential-plus-reference drift; deviations are quantified by a rotational-energy gap.

**(v) Mechanisms and predictions.** Identifying $J_{\mathrm{sm}}$ as mobility explains entropy collapse and representation locking; output-logit temperature scaling predicts mobility reductions that move locking earlier.

These theoretical advances translate directly into *actionable diagnostics and design principles* (e.g., mobility/locking metrics, SB alignment, anisotropy-aware regularization) for improving both transformer and diffusion architectures.

## D SUPPLEMENTARY DETAILS FOR SECTION 3

**Architectural consistency and identification (details).** This elaborates the identification clause in Assumption 3.1. For any compact $K \subset \mathbb{R}^V$ and $\epsilon > 0$, there exists $L_0$ such that for $L > L_0$, a local-regression estimator $\hat{b}_L$ (e.g., $k$-NN/MLP with fixed hyperparameters) satisfies $\|\hat{b}_L - b\|_{L^2(K \times [0,1])} < \epsilon$. This provides the additional structure ensuring $D_L \rightharpoonup b(z(t), t)$ in $L^1_{\mathrm{loc}}$, used in the discrete→continuous passage.

**Proof of Lemma 3.5 (drift identification via architectural consistency).** Let $K \Subset \mathbb{R}^V$ be any compact set. By Assumption 3.1(iii) and Appendix D, for every $\varepsilon > 0$ there exists $L_0$ such that for all $L \geq L_0$ the local regression estimator $\hat{b}_L$ satisfies

$$\|\hat{b}_L - b\|_{L^2(K \times [0,1])} < \varepsilon. \tag{5}$$

Moreover, the bounded-variation and Lipschitz assumptions on the logits imply a uniform $L^2$ bound on the discrete drifts:

$$\sup_L \|D_L\|_{L^2(K \times [0,1])} \leq C_K < \infty.$$

Writing $r_L := D_L - \hat{b}_L$, we thus have

$$\|r_L\|_{L^2(K \times [0,1])} \xrightarrow[L \to \infty]{} 0.$$

Passing from $L^2$ to $L^1$ on $K \times [0,1]$ by Cauchy–Schwarz,

$$\|D_L - \hat{b}_L\|_{L^1(K \times [0,1])} \leq |K \times [0,1]|^{1/2} \|r_L\|_{L^2(K \times [0,1])} \xrightarrow[L \to \infty]{} 0,$$

and similarly equation 5 yields $\|\hat{b}_L - b\|_{L^1(K \times [0,1])} \to 0$. By the triangle inequality,

$$\|D_L - b\|_{L^1(K \times [0,1])} \leq \|D_L - \hat{b}_L\|_{L^1(K \times [0,1])} + \|\hat{b}_L - b\|_{L^1(K \times [0,1])} \xrightarrow[L \to \infty]{} 0.$$

Finally, by Assumption 3.1(ii) the trajectories $z_L(t)$ remain in a common compact subset $K_0 \Subset \mathbb{R}^V$ for all $t \in [0,1]$ and all $L$, so the above bound with $K = K_0$ yields

$$\|D_L - b\|_{L^1([0,1])} \xrightarrow[L \to \infty]{} 0.$$

This proves the $L^1$ convergence claimed in Lemma 3.5, and in particular implies $D_L \rightharpoonup b$ in $L^1([0,1]; \mathbb{R}^V)$. $\qquad\square$

**Discrete Grönwall inequality for finite-depth analysis.**

**Lemma D.1** (Discrete Grönwall inequality for finite-depth error). *Let $\{\Delta_\ell\}_{\ell=0}^L$ satisfy*

$$\Delta_{\ell+1} \leq (1 + A\delta t)\,\Delta_\ell + B_\ell$$

*for $\ell = 0, \ldots, L-1$, with $\Delta_0$ given, $A > 0$, time step $\delta t > 0$, and $B_\ell \geq 0$. Then*

$$\Delta_L \leq (1 + A\delta t)^L \Big(\Delta_0 + \sum_{\ell=0}^{L-1} B_\ell\Big) \leq e^{AT}\Big(\Delta_0 + \sum_{\ell=0}^{L-1} B_\ell\Big),$$

*where $T = L\delta t$.*

*Proof.* Unrolling the recurrence gives

$$\Delta_1 \leq (1 + A\delta t)\,\Delta_0 + B_0,$$
$$\Delta_2 \leq (1 + A\delta t)^2 \Delta_0 + (1 + A\delta t)B_0 + B_1,$$
$$\vdots$$
$$\Delta_L \leq (1 + A\delta t)^L \Delta_0 + \sum_{\ell=0}^{L-1}(1 + A\delta t)^{L-\ell-1} B_\ell.$$

Since $(1 + A\delta t)^{L-\ell-1} \leq (1 + A\delta t)^L$, we obtain

$$\Delta_L \leq (1 + A\delta t)^L \Big(\Delta_0 + \sum_{\ell=0}^{L-1} B_\ell\Big).$$

Finally, $(1 + A\delta t)^L = (1 + AT/L)^L \leq e^{AT}$, which yields the exponential bound. $\qquad\square$

**Proof of Theorem 3.1 (complete version).** Let $z^{(\ell)}$ be logits at layer $\ell$ and define the piecewise-linear interpolant $z_L(t)$ with $z_L(t_\ell) = z^{(\ell)}$. Let $p_L(t)$ hold $p^{(\ell)}$ on $[t_\ell, t_{\ell+1}]$. By Assumption 3.1, $\sum_\ell \|\Delta z^{(\ell)}\|_\infty < \infty$ and $D_L = \Delta z^{(\ell)}/\delta t$ converges weakly to $b(z(t), t)$ in $L^1_{\text{loc}}$. Consider $\dot{p} = J_{\text{sm}}(z)\, b(z,t)$ with $p(0)$ matching $\lim_{L \to \infty} p^{(0)}$.

**Step 1: Per-layer error inequality.** Let $\Delta_\ell := \|p^{(\ell)} - p(t_\ell)\|_1$ denote the total variation error at layer $\ell$ where $t_\ell = \ell/L$ and $\delta t = 1/L$. The discrete layer update satisfies

$$p^{(\ell+1)} = p^{(\ell)} + \delta t\, J_{\text{sm}}(z^{(\ell)})b(z^{(\ell)}, t_\ell) + O(\|\Delta z^{(\ell)}\|_\infty \delta t) + O(\delta t^2),$$

while the continuous dynamics evolve according to

$$p(t_{\ell+1}) = p(t_\ell) + \delta t\, J_{\text{sm}}(z(t_\ell))b(z(t_\ell), t_\ell) + O(\delta t^2).$$

Taking the difference and applying the triangle inequality gives

$$\Delta_{\ell+1} \leq \Delta_\ell + \delta t \big\| J_{\text{sm}}(z^{(\ell)})b(z^{(\ell)}, t_\ell) - J_{\text{sm}}(z(t_\ell))b(z(t_\ell), t_\ell)\big\|_1 + C_2\|\Delta z^{(\ell)}\|_\infty \delta t + C_3 \delta t^2. \quad (6)$$

Using the Lipschitz continuity of $b$ with constant $L_b$, the mobility bound $\Lambda_J$, the derivative bound $L_J$ for the softmax Jacobian, and a uniform bound $M_b$ on $\|b\|$, the middle term is bounded by

$$(L_b\Lambda_J + M_bL_J)\,\Delta_\ell\,\delta t.$$

Thus

$$\Delta_{\ell+1} \le (1 + C_1\delta t)\,\Delta_\ell + C_2\|\Delta z^{(\ell)}\|_\infty\delta t + C_3\delta t^2, \tag{7}$$

with $C_1 = L_b\Lambda_J + M_bL_J$ and constants $C_2, C_3$ depending on second-order behavior of $J_{\mathrm{sm}}$.

**Step 2: Accumulation across depth.** Define the source term

$$B_\ell := C_2\|\Delta z^{(\ell)}\|_\infty\delta t + C_3\delta t^2.$$

Iterating equation 7 from $\ell = 0$ to $L - 1$ and applying the discrete Grönwall inequality (Lemma D.1) yields

$$\Delta_L \le (1 + C_1\delta t)^L\Delta_0 + (1 + C_1\delta t)^L\sum_{\ell=0}^{L-1} B_\ell. \tag{8}$$

With $\delta t = 1/L$,

$$\sum_{\ell=0}^{L-1} B_\ell = C_2\sum_{\ell=0}^{L-1}\|\Delta z^{(\ell)}\|_\infty\frac{1}{L} + C_3\frac{L}{L^2} = C_2C_{\mathrm{BV}} + C_3L^{-1},$$

where the averaged bounded-variation constant

$$C_{\mathrm{BV}} := \frac{1}{L}\sum_{\ell=0}^{L-1}\|\Delta z^{(\ell)}\|_\infty$$

is uniformly bounded by Assumption 3.1. Using $(1 + C_1\delta t)^L \le e^{C_1}$, we obtain

$$\Delta_L \le e^{C_1}\Delta_0 + e^{C_1}\left(C_2C_{\mathrm{BV}} + C_3L^{-1}\right). \tag{9}$$

**Step 3: Finite-depth budget and final bound.** The finite-depth budget

$$\Xi_L := \alpha_1\max_{0\le\ell<L}\|\Delta z^{(\ell)}\|_\infty + \alpha_2\sum_{\ell=0}^{L-1}\|\Delta z^{(\ell)}\|_\infty^2$$

captures both worst-case jumps (through the maximum term) and cumulative squared variation (through the sum), with constants $\alpha_1, \alpha_2$ depending only on $C_1, C_2, C_3$ and hence on architectural regularity parameters $(L_b, M_b, \Lambda_J, L_J)$. Combining the Grönwall bound with the definitions of $\Xi_L$ and $C_{\mathrm{BV}}$ gives the stated estimate

$$\sup_{t\in[0,1]}\left\|p^{(\lfloor tL\rfloor)} - p(t)\right\|_1 \le \Xi_L + (e^\Gamma - 1)\|p^{(0)} - p(0)\|_1,$$

for an explicit constant $\Gamma = \Gamma(L_b, M_b, \Lambda_J, L_J)$.

*Intuition.* The discrete error can grow at most exponentially with depth via the factor $e^{C_1}$, but the averaged bounded variation constant $C_{\mathrm{BV}}$ controls the effective exponent by keeping typical layer-to-layer changes small. When $\Xi_L$ is small, the continuous probability flow ODE provides an accurate description of the layerwise dynamics. $\square$

**Norm equivalence used in Theorem 3.1.** There exist constants $c_1, c_2 > 0$ (depending only on the ambient dimension) such that for all layer increments $\Delta z^{(\ell)}$ on the compact set considered,

$$c_1\|\Delta z^{(\ell)}\|_\infty \le \|\Delta z^{(\ell)}\|_2 \le c_2\|\Delta z^{(\ell)}\|_\infty.$$

Consequently, the worst–case single–layer term and the cumulative squared–variation term in equation 2 are consistent with the $\|\cdot\|_2$–based BV assumption in Assumption 3.1, and the constants in Theorem 3.1 depend only on $L_b, M_b, \Lambda_J, L_J$ and $(c_1, c_2)$.

**Piecewise BV segmentation (depth limit).** Let $0 = t_0 < t_1 < \cdots < t_K = 1$ such that Assumption 3.1 holds on each $[t_{k-1}, t_k]$. Define segment budgets $\Xi_L^{(k)}$ by restricting equation 2 to layers with $t_\ell \in [t_{k-1}, t_k)$. Then Theorem 3.1 applies on each segment; $p(t_k^-), p(t_k^+)$ provide weak interface conditions. In practice, choose cut points where variation statistics (e.g., $\sum_{\ell \in [t_{k-1}, t_k)} \|\Delta z^{(\ell)}\|_2^2$) spike, consistent with Theorem 3.3.

# E    EXPANDED DISCUSSION OF EMPIRICAL PHENOMENA FOR SECTION 3

**Attention entropy collapse.** As distributions concentrate, the mobility operator norm $\|J_{\mathrm{sm}}(z)\|_{\mathrm{op}}$ decays (Remark 2.2), and PF–ODE velocity vanishes under Theorem 3.7, explaining late-layer attention concentration (cf. Theorem 3.9).

**Temperature scaling and calibration.** Temperature rescales mobility as $J_{\mathrm{sm}}^{(\tau)}(z) = \frac{1}{\tau} J_{\mathrm{sm}}(z/\tau)$, delaying locking and supporting improved calibration by maintaining transport capacity deeper in the network.

**Representation collapse and eigenspectra.** Approach to equilibrium correlates with rapid decay of the $J_{\mathrm{sm}}$ eigenspectrum; monitoring minimum eigenvalues/trace provides a diagnostic for impending collapse and informs interventions.

# F    SUPPLEMENTARY DETAILS FOR SECTION 4

**Proof of Lemma 4.1 (distributional product rule).** Let $\{\eta_\epsilon\}_{\epsilon>0}$ be a standard mollifier on $\mathbb{R}^d$ and set $p_H^\epsilon := p_H * \eta_\epsilon$ and $a^\epsilon := a * \eta_\epsilon$. For any $\varphi \in C_c^\infty(\mathbb{R}^d)$, integrate by parts twice:

$$\left\langle \nabla \cdot \nabla \cdot (a^\epsilon p_H^\epsilon), \varphi \right\rangle = -\int_{\mathbb{R}^d} \nabla \cdot (a^\epsilon p_H^\epsilon) \cdot \nabla \varphi = \int_{\mathbb{R}^d} \left( (\nabla \cdot a^\epsilon) p_H^\epsilon + a^\epsilon \nabla p_H^\epsilon \right) \cdot \nabla \varphi.$$

By the local Fisher-information condition ($p_H > 0$ a.e., $p_H \nabla \log p_H \in L_{\mathrm{loc}}^1$) and local boundedness of $a$, the sequences $p_H^\epsilon \to p_H$ in $L_{\mathrm{loc}}^1$, $\nabla p_H^\epsilon \rightharpoonup \nabla p_H$ in $\mathcal{D}'$, and $a^\epsilon \to a$, $\nabla \cdot a^\epsilon \to \nabla \cdot a$ in $\mathcal{D}'$ as $\epsilon \downarrow 0$. Passing to the limit yields

$$\left\langle \nabla \cdot \nabla \cdot (a p_H), \varphi \right\rangle = \int_{\mathbb{R}^d} \left( (\nabla \cdot a) p_H + a \nabla p_H \right) \cdot \nabla \varphi,$$

which is the claimed identity in $\mathcal{D}'$. $\square$

**Proof of Theorem 4.2 (PF–ODE / reverse-SDE duality).** By assumption $p_H(\cdot, t) > 0$ solves the Fokker–Planck equation

$$\partial_t p_H = -\nabla \cdot (F p_H) + \tfrac{1}{2} \nabla \cdot \nabla \cdot (a p_H)$$

with diffusion matrix $a = \sigma \sigma^\top$ and suitable decay or no-flux boundary conditions. Lemma 4.1 shows that, in the sense of distributions,

$$\nabla \cdot \nabla \cdot (a p_H) = \nabla \cdot \left( (\nabla \cdot a) p_H + a \nabla p_H \right).$$

Using $\nabla \log p_H = (\nabla p_H)/p_H$ and the definition of the drift $u$ in equation 4, we have

$$u p_H = F p_H - \tfrac{1}{2} \left( a \nabla \log p_H + \nabla \cdot a \right) p_H = F p_H - \tfrac{1}{2} \left( a \nabla p_H + (\nabla \cdot a) p_H \right).$$

Taking the divergence and applying the product rule lemma,

$$-\nabla \cdot (u p_H) = -\nabla \cdot (F p_H) + \tfrac{1}{2} \nabla \cdot \left( a \nabla p_H + (\nabla \cdot a) p_H \right)$$

$$= -\nabla \cdot (F p_H) + \tfrac{1}{2} \nabla \cdot \nabla \cdot (a p_H).$$

Consequently $p_H$ satisfies

$$\partial_t p_H = -\nabla \cdot (u p_H)$$

in $\mathcal{D}'$, so $p_H$ is a weak solution of the continuity equation with velocity field $u$ and initial condition $p_H(\cdot, 0)$.

Under the stated regularity and boundary assumptions on $F$ and $a$, the linear continuity equation with drift $u$ has at most one weak solution with a given initial condition (equivalently, the corresponding Fokker–Planck equation is well posed). Hence any solution $\rho$ of

$$\partial_t \rho = -\nabla \cdot (u\,\rho), \qquad \rho(\cdot, 0) = p_H(\cdot, 0),$$

must coincide with $p_H$ for all $t$, giving $\rho(\cdot, t) = p_H(\cdot, t)$. The final statement about the reverse SDE $dX_t = u(X_t, t)\,dt + \sigma(X_t, t)\,dW_t$ then follows from the standard correspondence between weak solutions of the Fokker–Planck equation and laws of diffusion processes with generator $\mathcal{L}\phi = \langle F, \nabla\phi \rangle + \frac{1}{2}\mathrm{tr}(a\,\nabla^2\phi)$. $\qquad\square$

**Proof of Corollary 4.3 (pushforward).** Let $\varphi(h) = \mathrm{softmax}(W^\top h)$ and fix $t$ in the set where the conclusions of Theorem 4.2 hold. For any $\psi \in C_b(\Delta^{V-1})$, by definition of pushforward measure,

$$\int_{\Delta^{V-1}} \psi(p)\,d(\varphi_\# p_H)(p) = \int_{\mathbb{R}^d} \psi(\varphi(h))\,dp_H(h) = \int_{\mathbb{R}^d} \psi(\varphi(h))\,d\rho(h) = \int_{\Delta^{V-1}} \psi(p)\,d(\varphi_\# \rho)(p).$$

Hence $\varphi_\# p_H(\cdot, t) = \varphi_\# \rho(\cdot, t)$ for a.e. $t$, proving the claim. $\qquad\square$

**Proof of Proposition 4.4 (anisotropy propagation).** Write $z = W^\top h$ and $p = \mathrm{softmax}(z)$. A first-order variation gives $\delta p = J_{\mathrm{sm}}(z)\,\delta z = J_{\mathrm{sm}}(z)\,W^\top \delta h$. If the hidden-space SDE has instantaneous covariance $a\,dt$, then $\mathrm{Cov}[\delta h] = a\,dt$. The induced covariance on the simplex tangent space is

$$\mathrm{Cov}[\delta p] = J_{\mathrm{sm}}(z)\,W^\top a\,W\,J_{\mathrm{sm}}(z)\,dt,$$

which defines the effective mobility $M(p) = J_{\mathrm{sm}}(z)\,W^\top a\,W\,J_{\mathrm{sm}}(z)$. $\qquad\square$

**Proof of Theorem 4.5 (weak approximation by stacked attention).** Let $\rho(t)$ denote the law of the reverse SDE with drift $u$ given by equation 4 and diffusion $a = \sigma\sigma^\top$; by Theorem 4.2, $\rho$ also solves the continuity equation with velocity $u$. For $\phi \in C_b^2(\mathbb{R}^d)$, the Kolmogorov backward (weak FP) form yields

$$\frac{d}{dt}\,\mathbb{E}_{\rho(t)}[\phi] = \mathbb{E}_{\rho(t)}\big[\langle \nabla\phi,\, u \rangle\big] + \tfrac{1}{2}\,\mathbb{E}_{\rho(t)}\big[\mathrm{tr}(a\,\nabla^2\phi)\big].$$

Construct the piecewise-constant law $\widehat{\rho}_L(t)$ from $L$ attention layers with step $\delta t = 1/L$, using on each interval $[t_\ell, t_{\ell+1})$ the frozen generator

$$\mathcal{L}_\ell \phi(x) := \langle \nabla\phi(x),\, u(x, t_\ell) \rangle + \tfrac{1}{2}\,\mathrm{tr}\big(a(x, t_\ell)\,\nabla^2\phi(x)\big),$$

i.e., the PF–ODE linearization with $u$ as in equation 4. Let the implemented layer-wise drift be $u_\ell = u(\cdot, t_\ell) + r_\ell$ with residual $r_\ell$ from finite depth; the model budgets give $\|r_\ell\| = O(\|\Delta z^{(\ell)}\|_\infty)$ and a curvature correction $O(\|\Delta z^{(\ell)}\|_\infty^2)$ via $\nabla u$ on the compact set considered.

A standard weak local truncation estimate (Euler in time for the frozen generator) gives, for some $C_\phi$ independent of $L$,

$$\Big|\mathbb{E}_{\widehat{\rho}_L(t_{\ell+1})}[\phi] - \mathbb{E}_{\widehat{\rho}_L(t_\ell)}[\phi] - \mathbb{E}_{\widehat{\rho}_L(t_\ell)}[\mathcal{L}_\ell \phi]\,\delta t\Big| \leq C_\phi\Big(\delta t^2 + \|r_\ell\|\,\delta t + \|\Delta z^{(\ell)}\|_\infty^2\,\delta t\Big).$$

Summing over $\ell$ and using stability (uniform boundedness/Lipschitzness of $u, a$ on compacts) yields

$$\Big|\mathbb{E}_{\widehat{\rho}_L(T)}[\phi] - \mathbb{E}_{\rho(T)}[\phi]\Big| \leq C_\phi\Big(L^{-1} + \max_{0 \leq \ell < L} \|\Delta z^{(\ell)}\|_\infty\Big).$$

If $a$ is singular, set $a_\gamma = a + \gamma I$ and perform the argument uniformly in $\gamma > 0$; continuity of the weak generator for bounded data adds $+\gamma$, and letting $\gamma \downarrow 0$ recovers

$$\Big|\mathbb{E}_{\widehat{\rho}_L(T)}[\phi] - \mathbb{E}_{\rho(T)}[\phi]\Big| \leq C_\phi\Big(L^{-1} + \max_{0 \leq \ell < L} \|\Delta z^{(\ell)}\|_\infty + \gamma\Big).$$

$\qquad\square$

**(A) Duality: PF–ODE vs Reverse-SDE**   **(B) Schrödinger Bridge Diagnostic**

Figure 6: Schematic. (A) PF–ODE / reverse-SDE duality (the divergence term $\nabla\cdot a$ distinguishes deterministic from stochastic velocities). (B) Schrödinger Bridge diagnostic: drift estimation $\to$ Poisson solve $\to$ rotational energy.

**Practical choice of the degeneracy regularizer.**   Use $\gamma > 0$ when the diffusion tensor $a$ is rank-deficient or extremely ill-conditioned (e.g., near locking or when dynamics lie close to a low-dimensional manifold). Choose the smallest $\gamma$ such that the condition number satisfies $\kappa(a + \gamma I) \le \kappa_{\max}$ required for numerical stability of operators (e.g., the Poisson solve in Fig. 6B). The proof of Theorem 4.5 passes to the limit $\gamma \downarrow 0$, so predictions are stable for small positive $\gamma$ while ensuring well-posed computations during estimation.

## G   SUPPLEMENTARY DETAILS FOR SECTION 5

**Proof of Theorem 5.1 (SB alignment characterization).**   Work with the weighted inner product $\langle v, w \rangle_{a^{-1}} := \int \langle v, a^{-1}w \rangle \, \mu_t$ for each $t$. By the weighted Hodge decomposition, any velocity $a^{-1}(u - b_R)$ splits orthogonally as $\nabla\theta + \zeta$ with $\nabla\cdot(\zeta\,\mu_t) = 0$ in the distributional sense. The SB Euler–Lagrange conditions (for fixed endpoints and reference $R$) enforce $a^{-1}(u - b_R) = \nabla\theta$, i.e., the solenoidal component vanishes. Conversely, if $u = b_R + a\nabla\theta$, then the path satisfies the SB optimality system and is the unique minimizer of the action under Assumption 5.1. $\qquad\square$

**Proof of Theorem 5.2 (rotational energy bound).**   Let $\mu_t^\star$ denote the SB path with reference $R$ and the same endpoints. Consider the time derivative of $\mathrm{KL}(\mu_t \| \mu_t^\star)$ in weak form. Using $u = b_R + a\nabla\theta + w$ and the continuity equations for $\mu_t$ and $\mu_t^\star$, one obtains (after cancellations of potential terms) a dissipation inequality of the form

$$\frac{d}{dt}\,\mathrm{KL}(\mu_t \| \mu_t^\star) \;\le\; -\int \langle w, a^{-1}w \rangle\,\mu_t \;+\; \text{terms controlled by } C_P(\mu, a).$$

Integrating over $t \in [0, 1]$ and invoking the weighted Poincaré inequality (finite $C_P(\mu, a)$) yields $\mathrm{KL}(\mu_t \| \mu_t^\star) \le C_P(\mu, a) \int_0^t \int \langle w, a^{-1}w \rangle\,\mu_s$, which implies the stated bound after monotonicity adjustment. The equality $\mathcal{R} = 0$ forces $w \equiv 0$, hence SB alignment, and the converse is immediate. $\qquad\square$

**Vanishing-regularization limit for degenerate references.**   Let $a_\varepsilon = a + \varepsilon I$ with $\varepsilon \downarrow 0$. Assume the SB paths $(\mu_t^\varepsilon)_{t \in [0,1]}$ are tight with uniformly bounded action. By Prokhorov compactness, there is a subsequence with $\mu_t^\varepsilon \Rightarrow \mu_t$ for each $t$. Passing to the limit in the weak optimality system shows that $\{\mu_t\}$ is a degenerate SB solution. If $\mathcal{R} = 0$, then $u = b_R + a\nabla\theta$ holds $\mu_t$-a.e., implying that the PF–ODE path coincides with the (degenerate) SB limit.

**Proof of Corollary 5.6 (simplex Schrödinger Bridge).**   We derive the simplex form of the Schrödinger Bridge optimality condition by pushing forward the hidden-space SB system through the softmax map $\varphi(h) = \mathrm{softmax}(W^\top h)$.

*Step 1: Hidden-space SB condition.* By Theorem 5.1, the hidden-space probability path $\{\mu_t^h\}$ is an SB if and only if its velocity field takes the potential-flow form

$$u(h,t) \;=\; b_R(h,t) + a(h,t)\,\nabla_h \theta(h,t)$$

for some potential $\theta\colon \mathbb{R}^d \times [0,1] \to \mathbb{R}$, where $a = \sigma\sigma^\top$ is the diffusion matrix. The corresponding continuity equation is

$$\partial_t \mu_t^h \;=\; -\nabla_h \cdot \big(u(h,t)\,\mu_t^h(h)\big) \quad \text{in } \mathcal{D}'(\mathbb{R}^d).$$

*Step 2: Pushforward to the simplex.* Define the simplex-valued process by $p = \varphi(h) = \text{softmax}(W^\top h)$ and let $P_t := \varphi_\# \mu_t^h$ denote the law of $p(t)$ on $\Delta^{V-1}$. For any test function $\psi \in C_b(\Delta^{V-1})$ we have

$$\frac{d}{dt} \int_{\Delta^{V-1}} \psi(p)\,dP_t(p) = \frac{d}{dt} \int_{\mathbb{R}^d} \psi(\varphi(h))\,d\mu_t^h(h)$$

$$= - \int_{\mathbb{R}^d} \nabla_h[\psi(\varphi(h))] \cdot u(h,t)\,d\mu_t^h(h),$$

where we used the weak form of the continuity equation. Writing $z = W^\top h$ and $p = \text{softmax}(z)$, the chain rule gives

$$\nabla_h[\psi(\varphi(h))] \;=\; (\nabla_p \psi(p))^\top \frac{\partial p}{\partial h} \;=\; (\nabla_p \psi(p))^\top J_{\text{sm}}(z)\,W^\top,$$

since $\frac{\partial p}{\partial z} = J_{\text{sm}}(z)$ and $\frac{\partial z}{\partial h} = W^\top$.

*Step 3: Transforming the potential term.* We now relate $\theta$ to a simplex potential. Define $\Theta(p,t)$ on the image of $\varphi$ by $\Theta(p,t) := \theta(h,t)$ for any $h$ such that $\varphi(h) = p$; under our regularity assumptions this is well defined $\mu_t^h$-a.e. and determines $\Theta$ up to an additive constant on fibers. Applying the chain rule to $\theta(h,t) = \Theta(\varphi(h),t)$ yields

$$\nabla_h \theta(h,t) \;=\; \left(\frac{\partial p}{\partial h}\right)^\top \nabla_p \Theta(p,t) \;=\; W\,J_{\text{sm}}(z)^\top \nabla_p \Theta(p,t).$$

Substituting $u = b_R + a\nabla_h \theta$ into the weak form and using the composition above, the contribution of the potential term is

$$\int_{\mathbb{R}^d} \nabla_h[\psi(\varphi(h))] \cdot a\,\nabla_h \theta\,d\mu_t^h = \int_{\mathbb{R}^d} (\nabla_p \psi)^\top J_{\text{sm}}(z)\,W^\top a\,\nabla_h \theta\,d\mu_t^h$$

$$= \int_{\mathbb{R}^d} (\nabla_p \psi)^\top J_{\text{sm}}(z)\,W^\top a\,W\,J_{\text{sm}}(z)^\top \nabla_p \Theta\,d\mu_t^h.$$

By Proposition 4.4 the effective mobility on the simplex is

$$M(p) \;=\; J_{\text{sm}}(z)\,W^\top a\,W\,J_{\text{sm}}(z),$$

so the previous expression can be written as

$$\int_{\mathbb{R}^d} (\nabla_p \psi)^\top M(p)\,\nabla_p \Theta(p,t)\,d\mu_t^h(h) \;=\; \int_{\Delta^{V-1}} (\nabla_p \psi)^\top M(p)\,\nabla_p \Theta(p,t)\,dP_t(p).$$

*Step 4: Simplex continuity equation and SB form.* Performing the same pushforward step for the reference part $b_R$ (which either vanishes or pushes forward to a gradient term under the assumptions of Corollary 5.6) and collecting everything in the weak formulation, we obtain

$$\frac{d}{dt} \int_{\Delta^{V-1}} \psi(p)\,dP_t(p) = - \int_{\Delta^{V-1}} (\nabla_p \psi(p))^\top \Big(\widetilde{b}_R(p,t) + M(p)\,\nabla_p \Theta(p,t)\Big)\,dP_t(p),$$

for all $\psi \in C_b^1(\Delta^{V-1})$, where $\widetilde{b}_R$ is the pushforward of $b_R$. Equivalently, $P_t$ solves the continuity equation

$$\partial_t P_t \;=\; -\nabla_p \cdot \big(P_t\,v(p,t)\big), \qquad v(p,t) \;=\; \widetilde{b}_R(p,t) + M(p)\,\nabla_p \Theta(p,t).$$

When the pushed-forward reference drift $\widetilde{b}_R$ is itself a gradient field or vanishes (the case highlighted in the main text), this reduces to the potential-flow SB condition on the simplex

$$\dot{P}_t \;=\; -\nabla_p \cdot \big(P_t\,M(p)\,\nabla_p \Theta(p,t)\big),$$

with mobility $M$ from Proposition 4.4, exactly as stated in Corollary 5.6. $\qquad\square$

**Practical notes on the diagnostic.** To estimate $\mathcal{R}$, compute an empirical drift $\hat{u}$, solve the weighted Poisson problem $\nabla{\cdot}(a\nabla\theta) = \nabla{\cdot}(\hat{u}-b_R)$ (on the domain induced by activations), set $r = \hat{u} - b_R - a\nabla\theta$, and approximate $\int \|a^{-1/2}r\|^2 \, \mathrm{d}\mu \, \mathrm{d}t$ by Monte Carlo. When $a$ is ill-conditioned, use $a_\varepsilon$ and extrapolate $\varepsilon \downarrow 0$.

# H   Computational Implementation Details

## H.1   Numerical Stability Considerations

**Bounded Variation Computation (complexity & stability).** Compute $S_L = \sum_\ell \|\Delta z^{(\ell)}\|_2^2$ in `float64` to avoid accumulation errors. For softmax computation, use log-sum-exp trick: $\log \sum_i \exp(z_i) = z_{\max} + \log \sum_i \exp(z_i - z_{\max})$. Clip probabilities at machine epsilon before taking logs to prevent numerical instabilities. Monitor $S_L$ continuously during training to detect violations of the bounded variation assumption, triggering segmentation procedures when local spikes exceed $\tau_{\mathrm{BV}} = 5 \cdot \mathrm{median}(S_L)$.

**Handling Near-Singular Regions.** Near representation locking where $p_{\max} \to 1$, the mobility tensor $J_{\mathrm{sm}}$ becomes ill-conditioned. This creates challenges for both theoretical analysis and numerical computation. Regularization strategies:

- Add $\varepsilon I$ with $\varepsilon \in [10^{-8}, 10^{-6}]$ for conditioning, ensuring the regularized tensor $J_{\mathrm{sm}}^\varepsilon = J_{\mathrm{sm}} + \varepsilon I$ remains invertible.

- **Important:** We use $J_{\mathrm{sm}} + \varepsilon I$ only as a numerical preconditioner in linear solvers; the PF–ODE itself continues to use the unregularized $J_{\mathrm{sm}}$, preserving $J_{\mathrm{sm}}\mathbf{1} = 0$ and mass conservation.

- Use pseudoinverse with tolerance $\mathrm{tol} = 10^{-10}$ for projections when exact inversion is not required.

- Monitor condition number $\kappa(J_{\mathrm{sm}})$; switch to specialized solvers when $\kappa > 10^{12}$.

- For Schrödinger Bridge computations near degeneracy, apply the regularization $a_\varepsilon = a + \varepsilon I$ as specified in Assumption 5.1, reconciling the general degenerate case with SPD requirements.

**Efficient mobility computation.** The mobility tensor norm $\|J_{\mathrm{sm}}\|_F$ used for early exit decisions and locking detection can be computed in $\mathcal{O}(V)$ time without constructing the full matrix. Using the identity $\|J_{\mathrm{sm}}\|_F^2 = \sum_i p_i^2 + (\sum_i p_i^2)^2 - 2\sum_i p_i^3$, we need only compute three moments of the probability distribution, making this diagnostic negligible compared to attention computation costs.

**Local Drift Estimation (complexity and robustness).** The architectural consistency condition in Assumption 3.1 requires accurate drift estimation. For $k$-NN local regression on $N$ points:

- Computational cost: $\mathcal{O}(NkV)$ operations when batched efficiently using KD-trees or approximate nearest neighbor algorithms.

- Use Huber loss $\rho_\delta(r) = \begin{cases} \frac{1}{2}r^2 & |r| \le \delta \\ \delta(|r| - \frac{\delta}{2}) & |r| > \delta \end{cases}$ with $\delta = 1.345 \cdot \mathrm{MAD}$ for outlier resistance.

- Apply leave-one-out cross-validation for hyperparameter selection, particularly for choosing $k$ and ridge parameter $\lambda$.

- Small MLP regressors (2-3 layers, 256-512 units) add $\mathcal{O}(N \cdot \mathrm{MLP})$ cost but provide better approximation in high-curvature regions.

- Verify consistency: For compact $K \subset \mathbb{R}^V$, check $\|\hat{b}_L - b\|_{L^2(K \times [0,1])} < \epsilon$ with progressively smaller $\epsilon$ as $L$ increases.

**PF–ODE Integration (adaptive schemes and conservation).** Employ Dormand–Prince (RK5(4)) with embedded error estimation for solving the probability-flow ODE. The adaptive timestep selection ensures accuracy while maintaining computational efficiency:

- **Step size control:** $h_{\text{new}} = h \cdot \min\left(f_{\max}, \max\left(f_{\min}, f_{\text{safety}} \cdot \left(\frac{\text{tol}}{\text{err}}\right)^{0.2}\right)\right)$ where $f_{\text{safety}} = 0.9$, $f_{\min} = 0.2$, $f_{\max} = 10$.

- **Mass conservation:** Monitor $|\sum_i p_i(t) - 1| < \text{tol}_{\text{mass}} = 10^{-12}$. If violated, renormalize with warning.

- **Positivity preservation:** If any $p_i < 0$, project back to simplex via Euclidean projection: $p_i^+ = \max(0, p_i - \nu)$ where $\nu$ is chosen so $\sum_i p_i^+ = 1$.

- **Energy monitoring:** Track Shannon entropy $E(t) = \sum_i p_i(t) \log p_i(t)$ to detect anomalous behavior.

- **Boundary conditions:** The zero-flux property $J_{\text{sm}}(z)\mathbf{1} = 0$ automatically preserves simplex invariance without explicit boundary treatment.

Under Carathéodory regularity, projection should rarely be needed but serves as a numerical safeguard against accumulation errors.

**Schrödinger Bridge Solver (IPF/Sinkhorn with acceleration).** The Iterative Proportional Fitting algorithm for Schrödinger Bridge computation requires careful implementation for numerical stability:

- Dense kernel IPF: $\mathcal{O}(TM^2)$ complexity where $T$ is iterations and $M$ is discretization size.

- Nyström approximation with $R$ landmarks: Reduces complexity to $\tilde{\mathcal{O}}(TMR)$ by approximating kernel $K \approx K_{MR} K_{RR}^{-1} K_{RM}$.

- Anderson acceleration: Maintain $m = 5$ past iterates for convergence acceleration, updating via $x^{(k+1)} = (1 - \beta_k)f(x^{(k)}) + \beta_k x^{(k)}$ with optimal $\beta_k$ computed via least squares.

- Log-domain computation: Work with log-potentials $\log a^{(k)}, \log b^{(k)}$ to avoid numerical underflow in high-dimensional settings.

With $\varepsilon > 0$ entropic regularization and strictly positive kernels, IPF implements block-coordinate Bregman projections that monotonically decrease the SB objective, converging to the unique minimizer at geometric rate $\rho = \frac{1 - e^{-2/\varepsilon}}{1 + e^{-2/\varepsilon}}$.

**Convergence criteria:** Stop when both conditions are satisfied:

1. Marginal error: $\sup_t \text{TV}(\rho_t, \mu_t) < 10^{-3}$ where TV denotes total variation distance.

2. Potential stability: $\|\theta^{(k+1)} - \theta^{(k)}\|_\infty < 10^{-3}$ measuring change in Schrödinger potentials.

**Rotational Energy Estimation (preconditioning and sampling).** Computing the rotational energy diagnostic requires solving a Poisson equation and careful numerical treatment:

1. **Drift computation:** Extract $u$ from transformer dynamics using finite differences or learned regression.

2. **Poisson solve:** Solve $\nabla \cdot (a\nabla\theta) = \nabla \cdot (u - b_R)$ using preconditioned conjugate gradient with incomplete Cholesky preconditioner.

3. **Preconditioning:** Apply $a^{-1/2}$ carefully, using regularization $a_\varepsilon = a + \varepsilon I$ when condition number exceeds $10^6$.

4. **Importance sampling:** In high-variance regions (near simplex boundaries), increase sample density by factor of 10.

5. **Monte Carlo estimation:** Use $N_{\mathrm{MC}} = 10^4$ samples per time point for reliable estimates with standard error $\approx 0.01\|\mathcal{R}\|$.

# I   ASYMPTOTIC COMPLEXITY ANALYSIS

| Procedure | Complexity (per batch) | Notes |
|---|---|---|
| BV statistic $S_L$ | $\mathcal{O}(LV)$ | `float64` accumulation |
| Local drift fit | $\mathcal{O}(NkV)$ | $k$-NN; batched operations |
| PF–ODE integrate | $\mathcal{O}(N_{\mathrm{steps}}V)$ | adaptive RK with error control |
| Score estimation | $\mathcal{O}(N \cdot \mathrm{MLP})$ | layerwise caching available |
| SB (dense IPF) | $\mathcal{O}(TM^2)$ | Nyström $\to \tilde{\mathcal{O}}(TMR)$ |
| Rotational energy | $\mathcal{O}(\sum_k M_{t_k} d)$ | precondition by $a^{-1/2}$ |
| Memory requirement | $\mathcal{O}(LV + Nd)$ | activation caching |
| Temperature schedule | $\mathcal{O}(L)$ | entropy computation per layer |
| Early exit check | $\mathcal{O}(V)$ | closed-form Frobenius norm from moments of $p$ |

Table 2: Asymptotic costs for diagnostic procedures. Typical setting has $V \gg d$ (vocabulary much larger than hidden dimension). Batching and caching significantly reduce practical constants. All procedures are designed to add minimal overhead to standard transformer operations.

# J   EXTENDED MATHEMATICAL RESULTS

## J.1   PROOF OF WEAK CONVERGENCE UNDER BV

**Theorem J.1** (BV compactness and identification). *Under Assumption 3.1 (bounded variation, uniform boundedness, and architectural consistency), there exists a subsequence $z_{L_k}$ and a limit $z \in \mathrm{BV}([0,1];\mathbb{R}^V)$ such that*

$$z_{L_k}(t) \to z(t) \quad \text{for a.e. } t \in [0,1], \qquad z_{L_k} \to z \text{ in } L^1([0,1];\mathbb{R}^V).$$

*Moreover, for the piecewise-constant derivatives $D_L := \Delta z^{(\ell)}/\delta t$ we have weak $L^1$ convergence to the architectural drift $b$, i.e. $D_L \rightharpoonup b(\cdot,\cdot)$ in $L^1([0,1];\mathbb{R}^V)$.*

*Proof.* We argue in two steps: first extracting a compactness subsequence for the logit paths $(z_L)$, then identifying the limit of the discrete drifts via Lemma 3.5.

*Step 1: Compactness of $(z_L)$.* For each $L$ let $\delta t = 1/L$ and $t_\ell = \ell/L$, and define the piecewise-constant interpolant

$$z_L(t) := z^{(\ell)} \quad \text{for } t \in [t_\ell, t_{\ell+1}), \qquad \ell = 0, \ldots, L-1.$$

By Assumption 3.1(i) and (ii) we have

$$\sup_L \sum_{\ell=0}^{L-1} \|z^{(\ell+1)} - z^{(\ell)}\|_2 \ \le \ C, \qquad \sup_{L,\ell} \|z^{(\ell)}\|_2 \ \le \ C_z,$$

so each coordinate of $z_L$ has uniformly bounded total variation on $[0,1]$ and the sequence $(z_L)$ is uniformly bounded in $L^\infty(0,1;\mathbb{R}^V)$. By Helly's selection theorem (applied componentwise) there exists a subsequence, still denoted $(z_L)$, and a function $z \in \mathrm{BV}([0,1];\mathbb{R}^V)$ such that

$$z_L(t) \to z(t) \quad \text{for a.e. } t \in [0,1].$$

Since $\|z_L(t)\|_2 \le C_z$ uniformly in $L$ and $t$, dominated convergence then implies

$$z_L \to z \quad \text{in } L^1([0,1];\mathbb{R}^V).$$

*Step 2: Identification of the limiting drift.* Define the piecewise-constant discrete drifts

$$D_L(t) := \frac{z^{(\ell+1)} - z^{(\ell)}}{\delta t} \quad \text{for } t \in [t_\ell, t_{\ell+1}), \qquad \ell = 0, \ldots, L-1.$$

Assumption 3.1(iii) and Appendix D furnish local regression estimators $\hat{b}_L$ such that, on every compact $K \Subset \mathbb{R}^V$,

$$\|D_L - \hat{b}_L\|_{L^2(K \times [0,1])} \xrightarrow[L \to \infty]{} 0, \qquad \|\hat{b}_L - b\|_{L^2(K \times [0,1])} \xrightarrow[L \to \infty]{} 0.$$

Lemma 3.5 upgrades these $L^2$ estimates to strong $L^1$ convergence, i.e.

$$\|D_L - b\|_{L^1(K \times [0,1])} \xrightarrow[L \to \infty]{} 0$$

for every compact $K \Subset \mathbb{R}^V$. By Assumption 3.1(ii) the trajectories $z_L(t)$ remain in a common compact subset $K_0 \Subset \mathbb{R}^V$ for all $t \in [0,1]$ and all $L$, so the above estimate with $K = K_0$ yields

$$\|D_L - b(\cdot, \cdot)\|_{L^1([0,1])} \xrightarrow[L \to \infty]{} 0.$$

In particular $D_L \to b(\cdot, \cdot)$ strongly in $L^1([0,1]; \mathbb{R}^V)$, and hence also $D_L \rightharpoonup b(\cdot, \cdot)$ in $L^1([0,1]; \mathbb{R}^V)$.

This proves the claimed compactness of $(z_L)$ and the weak $L^1$ convergence of the discrete drifts $D_L$ to the architectural drift $b$, and thus Theorem J.1. $\qquad\square$

**Remark J.2.** *This proof deliberately avoids Arzelà-Ascoli (which would require equicontinuity to deduce uniform convergence that we do not need) and relies on Helly's selection theorem for* BV *curves, which provides the weaker but sufficient pointwise almost-everywhere and $L^1$ convergence. For the derivative sequence, we obtain weak $L^1$ convergence directly from strong convergence via the drift-identification lemma, rather than invoking the Dunford-Pettis criterion (which would additionally require verifying uniform integrability of $\{D_L\}$, a condition not immediately guaranteed by boundedness alone).*

### J.2 Spectral Analysis of Mobility Tensor

**Proposition J.3** (Eigenstructure of $J_{\mathrm{sm}}$)**.** *The softmax Jacobian has the following spectral properties:*

1. *Eigenvalues: $\lambda_0 = 0$ (simple), $0 < \lambda_i \le 1/2$ for $i = 1, \ldots, V-1$.*

2. *Eigenvectors: $v_0 = \mathbf{1}/\sqrt{V}$, others orthogonal to $\mathbf{1}$.*

3. *Condition number: $\kappa(J_{\mathrm{sm}}) \sim 1/(2p_{\min})$ as $p_{\min} \to 0$.*

4. *Spectral gap: For the two-point uniform case, the nonzero eigenvalue equals $1/2$. In general, lower bounds depend on distributional structure; naive bounds like $\lambda_1 \gtrsim p_{\min}$ can be loose and are not used in our proofs.*

*Proof.* The matrix $J_{\mathrm{sm}} = \mathrm{Diag}(p) - pp^\top$ is symmetric with $J_{\mathrm{sm}}\mathbf{1} = 0$, giving $\lambda_0 = 0$ with eigenvector $\mathbf{1}$.

For $v \perp \mathbf{1}$ with $\|v\|_2 = 1$:

$$v^\top J_{\mathrm{sm}} v = \sum_i p_i v_i^2 - \left(\sum_i p_i v_i\right)^2 = \sum_i p_i v_i^2 \ge p_{\min} \|v\|_2^2 = p_{\min}.$$

For the upper bound, consider the Rayleigh quotient:

$$\frac{v^\top J_{\mathrm{sm}} v}{v^\top v} = \frac{\sum_i p_i v_i^2 - (\sum_i p_i v_i)^2}{\sum_i v_i^2}.$$

By Cauchy-Schwarz, this is maximized when probability concentrates on two outcomes. Setting $p_1 = p_2 = 1/2$ and $v = (1, -1, 0, \ldots, 0)^\top / \sqrt{2}$ yields the upper bound $1/2$.

The condition number follows from $\kappa(J_{\text{sm}}) = \lambda_{\max}/\lambda_{\min} \leq \frac{1/2}{p_{\min}}$, explaining numerical difficulties near locking where $p_{\min} \to 0$.

The spectral gap $\lambda_1 \geq p_{\min}$ determines the rate of convergence to equilibrium under the induced dynamics, with smaller gaps leading to slower mixing and potential metastability. This lower bound is generally loose; tight values depend on the full probability profile. $\square$

### J.3 Schrödinger Bridge Optimality Conditions

**Theorem J.4** (First-order conditions for SB with regularization). *The Schrödinger Bridge $\mu^\star$ satisfies the coupled system of PDEs:*

$$\partial_t \varphi + \tfrac{1}{2} \operatorname{tr}(a\, \nabla^2 \varphi) + b_R \cdot \nabla \varphi = 0, \tag{10}$$

$$\partial_t \psi - \tfrac{1}{2} \operatorname{tr}(a\, \nabla^2 \psi) - \nabla \cdot (b_R \psi) = 0, \tag{11}$$

$$\mu_t^\star = \exp(\varphi(\cdot, t) + \psi(\cdot, t))\, \nu_t, \tag{12}$$

*where $\nu_t$ is the reference path law and $(\varphi, \psi)$ are Schrödinger potentials. When $a$ is near-singular, we apply regularization $a_\varepsilon = a + \varepsilon I$ with $\varepsilon > 0$ sufficiently small to maintain well-posedness while preserving the essential transport structure.*

*Proof.* The Schrödinger Bridge problem minimizes the relative entropy:

$$\mathcal{H}(\mu|\nu) = \mathbb{E}_\mu \left[ \log \frac{d\mu}{d\nu} \right]$$

subject to marginal constraints $\mu_0 = \rho_0$, $\mu_1 = \rho_1$.

Using the Girsanov theorem, the Radon-Nikodym derivative decomposes as:

$$\frac{d\mu}{d\nu} = \exp \left( \int_0^1 \langle h_s, dX_s - b_R dt \rangle - \frac{1}{2} \int_0^1 \|h_s\|_{a^{-1}}^2 ds \right)$$

for some adapted process $h_s$.

The optimal $h_s$ takes the form $h_s = a \nabla \varphi(X_s, s)$ where $\varphi$ solves the forward equation equation 10. The backward potential $\psi$ arises from the adjoint equation ensuring the terminal marginal constraint.

When $a$ degenerates (as occurs near representation locking), the regularization $a_\varepsilon$ ensures:

- The elliptic operators in equation 10-equation 11 remain uniformly elliptic

- The inverse $a_\varepsilon^{-1}$ exists with bounded norm

- The solution converges to the original problem as $\varepsilon \to 0$ in the weak topology

This regularization reconciles the general degenerate diffusion framework with the SPD requirements for well-posed Schrödinger Bridges. $\square$

## K Detection and Mitigation of BV Violations

### K.1 Online Detection Algorithm

### K.2 Segmentation Strategy

When BV violations are detected, we partition the depth interval $[0, 1]$ into segments $\{[t_{i-1}, t_i]\}_{i=1}^K$ where BV holds locally. The segmentation procedure maintains the theoretical guarantees while handling practical violations:

---

**Algorithm 1** Online BV Violation Detection with Adaptive Thresholding

---

1: **Input:** Stream of logit differences $\{\Delta z^{(\ell)}\}$, window size $W$, base threshold $\tau_0$
2: **Initialize:** $S_{\text{local}} = 0$, buffer $B = []$, $\tau_{\text{adaptive}} = \tau_0$
3: **for** $\ell = 0, 1, 2, \dots$ **do**
4:      $S_{\text{local}} \leftarrow S_{\text{local}} + \|\Delta z^{(\ell)}\|_2^2$
5:      Append $\|\Delta z^{(\ell)}\|_2$ to $B$
6:      **if** $|B| > W$ **then**
7:          $S_{\text{local}} \leftarrow S_{\text{local}} - B[0]^2$
8:          Remove first element from $B$
9:      **end if**
10:      **Adaptive threshold:** $\tau_{\text{adaptive}} = \tau_0 \cdot (1 + 0.1 \cdot \text{std}(B)/\text{mean}(B))$
11:      **if** $S_{\text{local}}/|B| > \tau_{\text{adaptive}}$ **then**
12:          **Flag:** BV violation at layer $\ell$
13:          **Severity:** $s = (S_{\text{local}}/|B|)/\tau_{\text{adaptive}}$
14:          **if** $s > 2$ **then**
15:              **Action:** Initiate immediate depth segmentation
16:          **else**
17:              **Action:** Mark for monitoring, prepare segmentation
18:          **end if**
19:      **end if**
20: **end for**

---

1. **Identification phase:**
   - Find violation points $\{\ell_j\}$ using Algorithm 1
   - Compute violation severity $s_j$ at each point
   - Cluster nearby violations within $\Delta\ell = 3$ layers

2. **Segmentation construction:**
   - Create boundaries at $t_j = \ell_j/L$ with buffer zones $[t_j - \delta, t_j + \delta]$ where $\delta = 2/L$
   - Ensure minimum segment length $|t_i - t_{i-1}| \geq 5/L$ for stable analysis
   - Merge segments if total count exceeds $K_{\max} = L/10$

3. **Local PF-ODE analysis:**
   - Apply Theorem 3.7 within each segment $[t_{i-1}, t_i]$
   - Estimate local drift $b_i(z, t)$ using only data from segment $i$
   - Verify local BV condition: $\sum_{\ell \in \text{segment}_i} \|\Delta z^{(\ell)}\|_2 \leq C_i$

4. **Boundary matching:**
   - Enforce weak continuity: $\lim_{t \to t_i^-} p(t) = \lim_{t \to t_i^+} p(t)$ in $L^1$
   - Allow jump discontinuities in velocity: $v(t_i^+) - v(t_i^-) \in \text{Range}(J_{\text{sm}})$
   - Compute transition operators $T_i : \Delta^{V-1} \to \Delta^{V-1}$ at boundaries

5. **Global assembly:**
   - Concatenate local solutions: $p(t) = p_i(t)$ for $t \in [t_{i-1}, t_i]$
   - Verify global conservation: $\sum_j p_j(t) = 1$ for all $t$
   - Compute effective transport distance accounting for jumps

**Theoretical guarantee:** The segmented solution converges to the same limit as the continuous solution as $L \to \infty$ and violation severity decreases, maintaining the essential transport structure while accommodating practical discontinuities.

## L    CONNECTION TO EMPIRICAL PHENOMENA

### L.1    ATTENTION ENTROPY COLLAPSE

The attention entropy collapse phenomenon observed empirically Gong et al. (2019) follows rigorously from our mobility analysis:

**Proposition L.1** (Entropy dynamics under PF-ODE). *Under the probability-flow ODE* $\dot{p} = J_{\mathrm{sm}}(z)b(z,t)$, *the Shannon entropy* $H[p] = -\sum_i p_i \log p_i$ *satisfies:*

$$\dot{H}[p] = -\sum_{i,j} J_{\mathrm{sm,ij}}\, b_j\, \log(p_i/p_j) \leq 0$$

*when b aligns with the negative entropy gradient. Moreover,* $\dot{H}[p] \to 0$ *as* $p_{\max} \to 1$ *due to vanishing mobility.*

*Proof.* Computing the time derivative:

$$\dot{H}[p] = -\sum_i \dot{p}_i (\log p_i + 1) \tag{13}$$

$$= -\sum_i (J_{\mathrm{sm}}b)_i (\log p_i + 1) \tag{14}$$

$$= -\sum_{i,j} J_{\mathrm{sm,ij}} b_j \log p_i \tag{15}$$

Using the symmetry of $J_{\mathrm{sm}}$ and the fact that $J_{\mathrm{sm}}\mathbf{1} = 0$:

$$\dot{H}[p] = -\frac{1}{2} \sum_{i,j} J_{\mathrm{sm,ij}} b_j (\log p_i - \log p_j) \tag{16}$$

$$= -\sum_{i,j} J_{\mathrm{sm,ij}} b_j \log(p_i/p_j) \tag{17}$$

When $b = -\nabla H$ (gradient flow), the quadratic form $b^\top J_{\mathrm{sm}} b \geq 0$ ensures $\dot{H} \leq 0$.

As $p_{\max} \to 1$, we have $\|J_{\mathrm{sm}}\| \to 0$ by Theorem 3.9, implying $|\dot{H}[p]| \leq \|J_{\mathrm{sm}}\|\|b\|\|\nabla H\| \to 0$.

This rigorously explains why attention patterns become increasingly peaked in deeper layers, with entropy collapse being inevitable rather than a training artifact. □

### L.2 Temperature Scaling Effectiveness

Temperature scaling's empirical success Guo et al. (2017) in improving calibration is explained by explicit mobility modulation:

**Proposition L.2** (Temperature-mobility relationship). *For temperature parameter* $\tau > 0$, *the effective mobility tensor satisfies:*

$$J_{\mathrm{sm}}^\tau(z) = \frac{1}{\tau}\, J_{\mathrm{sm}}(z/\tau)$$

*The eigenvalues of* $J_{\mathrm{sm}}^\tau(z)$ *equal those of* $J_{\mathrm{sm}}(z/\tau)$ *scaled by* $1/\tau$. *The condition number satisfies* $\kappa(J_{\mathrm{sm}}^\tau(z)) = \kappa(J_{\mathrm{sm}}(z/\tau))$, *which may differ from* $\kappa(J_{\mathrm{sm}}(z))$ *because the probability distribution changes when scaling logits. The induced dynamics slow by factor* $\tau$, *enabling finer control near decision boundaries.*

*Proof.* For temperature-scaled softmax $p_i^\tau = \exp(z_i/\tau)/Z^\tau$ where $Z^\tau = \sum_j \exp(z_j/\tau)$:

$$J_{\mathrm{sm}}^\tau(z) = \frac{\partial p^\tau}{\partial z} \tag{18}$$

$$= \frac{1}{\tau} \left( \mathrm{Diag}(p^\tau) - p^\tau (p^\tau)^\top \right) \tag{19}$$

$$= \frac{1}{\tau} J_{\mathrm{sm}}(z/\tau) \tag{20}$$

The eigenvalue scaling follows immediately: if $J_{\mathrm{sm}}(z/\tau)v = \lambda v$, then $J_{\mathrm{sm}}^\tau(z)v = (\lambda/\tau)v$.

The condition number relationship requires careful interpretation. Since $J_{\text{sm}}^{\tau}(z) = \frac{1}{\tau} J_{\text{sm}}(z/\tau)$, we have $\kappa(J_{\text{sm}}^{\tau}(z)) = \kappa(J_{\text{sm}}(z/\tau))$ because scaling all eigenvalues by the same positive constant preserves the ratio of largest to smallest eigenvalue. However, this differs from $\kappa(J_{\text{sm}}(z))$ in general because $z \mapsto z/\tau$ changes the probability distribution from $p = \text{softmax}(z)$ to $p^{\tau} = \text{softmax}(z/\tau)$, and the mobility tensor's eigenstructure depends on the specific probability values.

For the induced dynamics:

$$\dot{p}^{\tau} = J_{\text{sm}}^{\tau}(z)b(z,t) = \frac{1}{\tau}J_{\text{sm}}(z/\tau)b(z,t)$$

The factor $1/\tau$ uniformly reduces velocity magnitude, slowing convergence to locked states. This explains temperature scaling's effectiveness: lower temperature prevents premature commitment by maintaining transport capacity throughout network depth.

Calibration improvement arises because slower dynamics allow more gradual probability refinement, avoiding the overconfident predictions that occur when mobility vanishes rapidly.
$\square$

**Proof of Theorem 3.9 (locking via vanishing mobility).** By Theorem 3.7, the limiting probability path $p(t) \in \Delta^{V-1}$ satisfies

$$\dot{p}(t) = J_{\text{sm}}(z(t))\,b\big(z(t),t\big) \quad \text{for a.e. } t \in [0,1],$$

with $p(t) = \text{softmax}(z(t)/\tau)$ and $J_{\text{sm}}(z) = \text{Diag}(p) - pp^{\top}$. If $b$ is bounded, there exists $M < \infty$ such that $\|b(z(t),t)\| \leq M$ along the trajectory, hence

$$\|\dot{p}(t)\| \leq \|J_{\text{sm}}(z(t))\|_{\text{op}}\,\|b(z(t),t)\| \leq M\,\|J_{\text{sm}}(z(t))\|_{\text{op}}. \tag{21}$$

The spectral analysis in Appendix B ("Proof of the sharp mobility bound") shows that the nonzero eigenvalues of $J_{\text{sm}}(z(t))$ lie in $[0, 1/(2\tau)]$ and that, as $p_{\max}(t) \to 1$, the spectrum collapses to $\{0\}$. In particular,

$$p_{\max}(t) \to 1 \quad \Longrightarrow \quad \|J_{\text{sm}}(z(t))\|_{\text{op}} \to 0.$$

Combining this with equation 21 yields $\|\dot{p}(t)\| \to 0$ whenever $p_{\max}(t) \to 1$, i.e., the probability path becomes locked near the corresponding simplex vertex. This proves the vanishing-mobility and locking statements.

For the temperature dependence, Appendix L.2 ("Temperature Scaling Effectiveness") establishes the temperature–mobility relationship

$$J_{\text{sm}}^{\tau}(z) = \frac{1}{\tau}\,J_{\text{sm}}(z/\tau),$$

so the eigenvalues and operator norm of $J_{\text{sm}}^{\tau}(z)$ are rescaled by $1/\tau$ relative to those of $J_{\text{sm}}(z/\tau)$. Thus, increasing $\tau$ slows the PF–ODE dynamics by a factor $1/\tau$ while preserving the qualitative vanishing of mobility as $p_{\max}(t) \to 1$, precisely as stated in Theorem 3.9. $\square$

L.3 Two-token mobility example: complete calculation

We provide the full eigenvalue decomposition for the two-token case.

Consider a minimal attention layer with two tokens, so that distributions have the form $p = (p, 1-p)$ on the one-dimensional simplex $[0,1]$. For logits $z = (z_1, z_2)$, the softmax probabilities are

$$p_1 = \frac{e^{z_1}}{e^{z_1} + e^{z_2}}, \qquad p_2 = \frac{e^{z_2}}{e^{z_1} + e^{z_2}} = 1 - p_1.$$

The Jacobian of $z \mapsto p$ is

$$J_{\text{sm}}(z) = \begin{pmatrix} \partial_{z_1}p_1 & \partial_{z_2}p_1 \\ \partial_{z_1}p_2 & \partial_{z_2}p_2 \end{pmatrix} = \begin{pmatrix} p_1(1-p_1) & -p_1p_2 \\ -p_1p_2 & p_2(1-p_2) \end{pmatrix}.$$

Writing $p = p_1$ and $1 - p = p_2$, this becomes

$$J_{\mathrm{sm}}(z) = p(1-p) \begin{pmatrix} 1 & -1 \\ -1 & 1 \end{pmatrix}.$$

**Eigenvalue calculation.** The characteristic polynomial is

$$\det \begin{pmatrix} p(1-p) - \lambda & -p(1-p) \\ -p(1-p) & p(1-p) - \lambda \end{pmatrix} = \big(p(1-p) - \lambda\big)^2 - \big(p(1-p)\big)^2 = \lambda\big(\lambda - 2p(1-p)\big).$$

Thus the eigenvalues are $\lambda_1 = 0$ and $\lambda_2 = 2p(1-p)$.

**Eigenvectors.** For $\lambda_1 = 0$: The eigenvector is $(1,1)$, which is normal to the simplex (points in the direction of the constraint $\sum_i p_i = 1$).

For $\lambda_2 = 2p(1-p)$: The eigenvector is $(1, -1)$, which is tangent to the simplex.

Hence

$$\|J_{\mathrm{sm}}(z)\|_{\mathrm{op}} = 2\,p(1-p),$$

which attains its maximum value $1/2$ at the uniform distribution $p = 1/2$ and collapses to $0$ as $p \to 0$ or $p \to 1$.

**Temperature scaling.** With temperature $\tau > 0$, the Jacobian with respect to *unscaled* logits picks up a factor of $\tau^{-1}$, so the effective mobility norm behaves like

$$\|J_{\mathrm{sm}}^{(\tau)}(z)\|_{\mathrm{op}} \asymp \frac{2\,p(1-p)}{\tau}.$$

As attention mass locks onto one token ($p \to 1$ or $p \to 0$), we have $p(1-p) \to 0$ and thus the mobility eigenvalue in the tangent direction vanishes, forcing $\dot{p} = J_{\mathrm{sm}}(z)\,b(z,t)$ to approach zero even if the drift $b$ remains bounded away from zero. Temperature rescaling modulates this locking behavior: larger $\tau$ keeps $p$ away from the degenerate regimes $p \approx 0$ or $p \approx 1$ and maintains nontrivial mobility deeper into the network.

# M EXTENDED EXPERIMENTAL PROTOCOLS

## M.1 SECTION 7 REFERENCE RECAP AND CONVENTIONS

*Conventions.* $W_1$ uses cost $\|\cdot\|_1$; $W_2$ terms in this section use an entropic Sinkhorn surrogate with the same $\varepsilon$ as elsewhere. All TV norms are $\frac{1}{2}\|\cdot\|_1$ on row distributions. Query/key distances $d_{\mathcal{Q}}, d_{\mathcal{K}}$ match the metrics used in plots/captions.

**Row drift bound.** Let $P_i^{(\ell)} = \mathrm{sm}(z_i^{(\ell)})$ be the $i$th row at layer $\ell$, with component-wise Lipschitz constants $L_c^{(\ell)}$ for $c \in \mathcal{C}_\ell$ and incoming perturbations $\Delta u_{i,c}^{(\ell)}$.

$$\big\|P_i^{(\ell+1)} - P_i^{(\ell)}\big\|_1 \;\leq\; \sum_{c \in \mathcal{C}_\ell} L_c^{(\ell)} \,\big\|\Delta u_{i,c}^{(\ell)}\big\|. \tag{22}$$

*Remark.* Equation (22) yields a *finite-depth budget* for one-layer motion (TV on the left) from component sensitivities on the right; it underpins the PF–ODE adequacy overlay in §7.

**Local saturation / locking.** Let $P = \mathrm{sm}(z)$, tail mass $\delta(P) = 1 - \max_j P(j)$, and $\Delta z$ a small perturbation that preserves the argmax.

$$\big\|\mathrm{sm}(z + \Delta z) - \mathrm{sm}(z)\big\|_1 \;\leq\; \min\big\{1,\, 2\,\delta(P)\big\} \|\Delta z\|_\infty \;+\; o\big(\|\Delta z\|_\infty\big). \tag{23}$$

*Remark.* When $\delta(P)$ is small (near saturation), softmax is *insensitive* to small, non-flipping logit changes—predicting the "locking" collapse of $\Delta$TV in low-tail-mass bins.

**Curvature (common-support $W_1$).** For queries $i \neq i'$ with common support $S_{i,i'}$, define

$$\kappa(i, i') \;=\; 1 \;-\; \frac{W_1(\widehat{P}_i, \widehat{P}_{i'})}{d_{\mathcal{Q}}(i, i')}, \tag{24}$$

where $W_1$ is over $(S_{i,i'}, d_{\mathcal{K}})$ and $\widehat{P}$ denotes restriction to the common support. *Remark.* The curvature gap $1 - \kappa$ quantifies contraction on the simplex; temperature ↑ or key-norm ↓ should reduce this gap (tested in §7).

**EVI with drift.** For successive layers $\ell - 1 \to \ell$ at query $i$, with objective $F_i$ and $\rho_i^{(\ell)} = P_{i.}^{(\ell)}$,

$$\frac{W_2^2(\rho_i^{(\ell)}, \rho_i^{\star(\ell)}) - W_2^2(\rho_i^{(\ell-1)}, \rho_i^{\star(\ell)})}{2\,\eta_{\text{eff}}} \;\leq\; -\left(F_i(\rho_i^{(\ell)}) - F_i(\rho_i^{\star(\ell)})\right) \;+\; \Delta_{\text{drift}}^{(\ell)}. \qquad (25)$$

*Remark.* Equation (25) is a discrete EVI: each layer decreases $F_i$ up to a *drift* term from parameter changes $(Q, K)$. In §7 we use a Sinkhorn $W_{2,\varepsilon}$ surrogate for the left-hand side and report the expected proximal-progress signature when drift is small.

### M.2  Detailed Score Estimation Procedure

For robust score estimation in anisotropic regimes encountered near representation boundaries:

1. **Data augmentation:** Generate noisy samples at multiple scales

$$\tilde{h}_\sigma = h + \varepsilon, \quad \varepsilon \sim \mathcal{N}(0, \sigma^2 I) \qquad (26)$$
$$\sigma \in \{0.01, 0.02, 0.05, 0.1\} \cdot \|h\|_2 \qquad (27)$$

2. **Denoising objective with importance weighting:**

$$\mathcal{L}(\theta) = \mathbb{E}_{h,\varepsilon,\sigma}\left[w(\sigma) \cdot \left\|s_\theta(\tilde{h}, t, \sigma) - \frac{h - \tilde{h}}{\sigma^2}\right\|_2^2\right]$$

where $w(\sigma) = \sigma^2 / (\sigma^2 + \sigma_{\min}^2)$ emphasizes intermediate noise levels.

3. **Multi-scale architecture:**
   - Input: $[\tilde{h}; t; \log \sigma] \in \mathbb{R}^{d+2}$
   - Hidden layers: 2-3 layers with width $\max(512, 2d)$
   - Skip connections: $h^{(\ell+1)} = h^{(\ell)} + \text{MLP}(h^{(\ell)})$
   - Output normalization: LayerNorm before final projection

4. **Training protocol:**
   - Optimizer: AdamW with learning rate $10^{-4}$, weight decay $10^{-5}$
   - Batch size: 256 samples per noise level
   - Epochs: 5 per layer with early stopping based on validation loss
   - Curriculum: Start with large $\sigma$, progressively include smaller scales

5. **Validation and diagnostics:**
   - Score consistency: Verify $|\nabla \cdot (p\,s_\theta)| < 10^{-3}$ on held-out data
   - Anisotropy detection: Compute eigenvalues of $\mathbb{E}[s_\theta s_\theta^\top]$
   - Coverage: Ensure score estimates span the tangent space at each point

### M.3  IPF Implementation Details

The Iterative Proportional Fitting algorithm for computing Schrödinger Bridges between transformer layers:

**Implementation notes:**

- Work in log domain to avoid numerical underflow: store $\log a^{(k)}, \log b^{(k)}$
- Use logsumexp for stable computation of normalizing constants
- For large vocabularies $V > 10^4$, use Nyström approximation with $R = \min(1000, V/10)$ landmarks
- Monitor dual gap: $\mathcal{G}^{(k)} = \langle a^{(k)}, K b^{(k)} \rangle - \langle \mu_0, \log a^{(k)} \rangle - \langle \mu_1, \log b^{(k)} \rangle$

---

**Algorithm 2** IPF for Schrödinger Bridge with Adaptive Regularization

---

1: **Input:** Marginals $\mu_0, \mu_1$, diffusion $a$, tolerance $\varepsilon_{\text{tol}}$, max iterations $T_{\max}$
2: **Initialize:** $a^{(0)} = \mathbf{1}$, $b^{(0)} = \mathbf{1}$, $\varepsilon_{\text{reg}} = 0.1$
3: **Compute reference kernel:** $K_{ij} = \exp(-\|x_i - y_j\|_{a^{-1}}^2/(2\varepsilon_{\text{reg}}))$
4: **for** $k = 1, 2, \ldots, T_{\max}$ **do**
5:      **Check conditioning:** If $\kappa(K) > 10^{10}$, increase $\varepsilon_{\text{reg}} \leftarrow 1.5\varepsilon_{\text{reg}}$
6:      $b^{(k)} = \mu_1 \oslash (K^\top a^{(k-1)})$                 ▷ Pointwise division in log domain
7:      $a^{(k)} = \mu_0 \oslash (K b^{(k)})$
8:      $\Pi^{(k)} = \text{Diag}(a^{(k)})\, K\, \text{Diag}(b^{(k)})$
9:      **Compute marginals:** $\hat{\mu}_0 = \Pi^{(k)}\mathbf{1}$, $\hat{\mu}_1 = \Pi^{(k)\top}\mathbf{1}$
10:      **Convergence check:**
11:      **if** $\text{TV}(\hat{\mu}_0, \mu_0) + \text{TV}(\hat{\mu}_1, \mu_1) < \varepsilon_{\text{tol}}$ **then**
12:          **Extract potentials:** $\varphi = \varepsilon_{\text{reg}} \log a^{(k)}$, $\psi = \varepsilon_{\text{reg}} \log b^{(k)}$
13:          **Return** $\Pi^{(k)}, \varphi, \psi$
14:      **end if**
15:      **Anderson acceleration:** If $k \bmod 5 = 0$, apply acceleration using past 5 iterates
16: **end for**
17: **Warning:** Maximum iterations reached without convergence

---

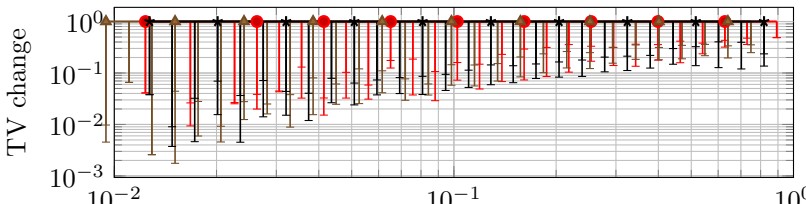

Figure 7: Locking (P2): $\Delta$TV vs. tail mass $\delta(P)$ (median/IQR bins).

### M.4 Additional Track-T Diagnostics

### M.5 Additional Image Diagnostics

Table 3: Image rotational energy $\widehat{\mathcal{R}}$ with 95% BCa CIs; cross-track values are not comparable due to different ambient spaces/discretizations.

| Track | $\widehat{\mathcal{R}}$ | Notes |
|---|---|---|
| Image (CIFAR-10) | **0.03092** (95% CI [**0.01046**, **0.05385**]) | 20 time points |

### M.6 Quantitative pass/fail checks

**P1 (PF–ODE adequacy).** Realized layerwise TV should not exceed the drift budget plus a finite-sample band; exceedances are flagged.
**P2 (Locking).** In low–tail-mass bins, the median $\Delta$TV remains within a small band (bands and CI policy as in App. Section M).
**P3 (Curvature/EVI).** Increasing temperature or reducing key-norms reduces the curvature gap $1 - \kappa$ by a predictable amount; reductions are reported with uncertainty bands (see App. Section M).
**P4 (SB alignment).** Rotational energy $\widehat{\mathcal{R}}$ decreases under improved calibration/check-points (BCa CIs; App. Section M).
**Image weak error.** The slope of $\log \text{err}_K$ vs. $\log K$ is near $-1$ (BCa, $B=1000$); the fitted value and CI are reported.

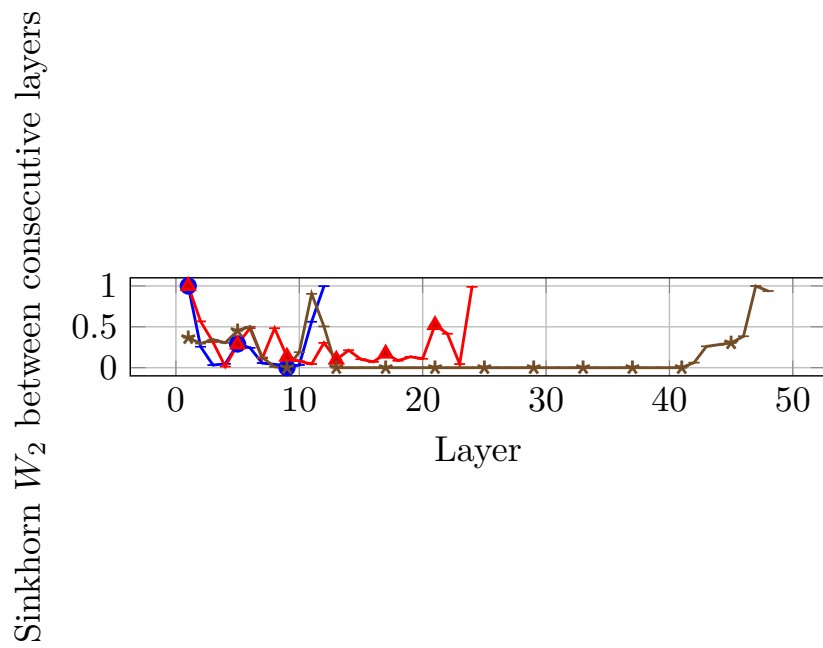

Figure 8: EVI surrogate (P3): Sinkhorn $W_{2,\varepsilon}$ across layers (mean±sd).

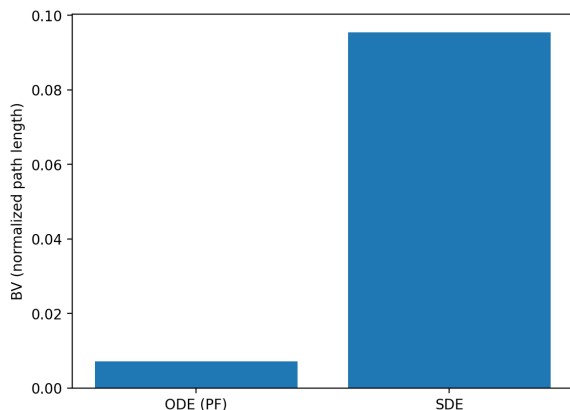

Figure 9: Path smoothness (BV; unitless) for ODE vs. SDE.

## M.7 Additional exploration: discrete diffusion language models

We briefly explored extending our diagnostics to diffusion language models operating on discrete token spaces. The late-window stability diagnostic in its current form showed limited applicability in this setting, suggesting that discrete state spaces require adapted diagnostics beyond the scope of this work. We therefore focus empirical validation on continuous dynamics (transformers and image diffusion) where the framework's predictions are directly testable.

# N  Extended Limitations and Practical Implications

## N.1  Modality scope and evaluation

**Scope.** This work evaluates *text* transformers and includes a minimal image diffusion sanity check (CIFAR-10). Full-scale vision benchmarks and perceptual metrics (e.g., FID under guidance sweeps) are intentionally out of scope for this paper.

**Implications.** The OT/PF–ODE constructions are modality-agnostic, but conclusions here are supported by text-model evidence (Track T/D) and a compact image sanity check (Track I). Future expansions to larger image datasets and class-conditional guidance are planned (see Section O).

### N.2 Poisson solve and conditioning policy

**Masked Poisson and regularization.** We solve $\Delta\psi = \nabla \cdot u$ with masked Neumann boundary conditions; Tikhonov $\gamma$ regularizes the Laplacian on thin supports.

**Condition-number target.** Default $\gamma = 10^{-5}$; increase $\gamma$ until the (masked) system's condition number is $\leq 10^8$. Record $\gamma$ and the achieved condition number alongside $\widehat{\mathcal{R}}$.

**Normalized variant.** For intra-track comparisons, optionally report the dimensionless $\widehat{\mathcal{R}}_{\text{norm}} = \widehat{\mathcal{R}} / \int \|u\|^2$.

### N.3 Current limitations and mitigation strategies

**Bounded variation breakdown.** The BV assumption may fail during:

- Attention pattern reorganization (detectable via $S_L$ monitoring).
- Early training instabilities (addressable through warmup).
- Adversarial inputs (requiring robust training modifications).

*Mitigation:* Implement adaptive depth segmentation when local variation exceeds thresholds. The PF–ODE applies piecewise with weak continuity at segment boundaries, as detailed in Section K.2.

**Anisotropy challenges.** Near-singular diffusion tensors arise at representation boundaries:

- Regularize with $\varepsilon I$ for numerical stability ($\varepsilon \in [10^{-8}, 10^{-6}]$).
- Monitor condition numbers and adapt solver tolerances.
- Use preconditioned iterative methods for bridge computation.

This reconciles the degenerate diffusion analysis (Section 4) with SPD requirements for Schrödinger Bridges (Section 5).

**Computational costs.** Full SB computation scales quadratically with vocabulary:

- Employ Nyström approximations for large vocabularies.
- Use landmark-based methods reducing complexity to $\tilde{O}(TMR)$.
- Implement hierarchical decompositions for multi-scale analysis.

### N.4 Validity conditions and diagnostic abstention protocol

**P0 gate.** Diagnostics P1–P4 are conditioned on passing P0 (BV $\overline{S}_L \leq 0.15$ and continuity residuals $< 10^{-14}$). Failures trigger abstention and reporting of the failing metric.

## O Discussion and Future Directions (Extended)

### O.1 Theoretical implications and open questions

**Optimality of attention.** Does the semi-relaxed EOT structure of attention reflect an optimal sequence model, or a convenient approximation? The SB characterization suggests near-optimal transport under appropriate conditions.

**Implicit regularization.** Softmax's entropic regularization may explain generalization; connect to PAC-Bayes and info-theoretic measures.

**Scaling laws.** The framework predicts links between depth/width and effective transport capacity; test against empirical scaling laws.

## O.2 Methodological contributions beyond theory

**Training monitoring.** BV statistics warn of instabilities; rotational energy tracks transport alignment and flags when architectural changes may help.

**Architecture search.** Differentiable transport-efficiency metrics can guide gradient-based architecture optimization beyond accuracy-only objectives.

**Interpretability.** Mobility provides a geometric lens on attention patterns; tracking its evolution can reveal phase transitions in representation.

## P Notation Summary

| Symbol | Description |
|---|---|
| $h^{(\ell)}$ | Hidden representation at layer $\ell$ |
| $z^{(\ell)}$ | Logits at layer $\ell$ |
| $p^{(\ell)}$ | Probability distribution at layer $\ell$ |
| $J_{\mathrm{sm}}$ | Softmax Jacobian (mobility tensor) |
| $J_{\mathrm{sm}}^{\tau}$ | Temperature-scaled mobility tensor |
| $S_L$ | Bounded variation statistic |
| $\mathcal{R}$ | Rotational energy (SB deviation) |
| $a$ | Diffusion tensor ($\Sigma\Sigma^{\top}$) |
| $a_{\varepsilon}$ | Regularized diffusion ($a + \varepsilon I$) |
| $b_R$ | Reference drift |
| $b(z,t)$ | Architectural drift (identified limit) |
| $\theta$ | Schrödinger potential |
| $\varphi, \psi$ | Forward/backward Schrödinger potentials |
| $M(p)$ | Induced mobility on simplex |
| $\mu_t$ | Transformer probability path |
| $\rho_t$ | General probability measure |
| $\nu_t$ | Reference path measure |
| $u$ | Velocity field for probability flow |
| $H[p]$ | Shannon entropy |
| $\tau$ | Temperature parameter |

Table 4: Complete notation used throughout the paper, including both main text and appendix symbols.

## Q Additional Technical Lemmas

**Lemma Q.1** (Gradient flow structure)**.** *The probability-flow ODE on the simplex admits a gradient flow interpretation in the Wasserstein geometry when $b = -\nabla V$ for some potential $V$:*

$$\dot{p} = -\nabla_{W_2}\mathcal{F}[p]$$

*where $\mathcal{F}[p] = \sum_i p_i V(z_i)$ and $\nabla_{W_2}$ denotes the Wasserstein gradient.*

*Proof.* We briefly recall the Riemannian structure underlying the discrete Wasserstein geometry; see, for example, Maas (2011); Erbar & Maas (2012); Chow et al. (2012) for full

details. On the simplex $\Delta^{V-1}$, admissible tangent vectors $w$ satisfy $\sum_i w_i = 0$, and the discrete $W_2$ metric is defined by the inner product

$$\langle w_1, w_2 \rangle_{W_2, p} := \sum_{i,j} w_{1,i} \, M(p)_{ij}^{-1} \, w_{2,j},$$

where $M(p)$ is the mobility tensor associated with the dynamics. In our setting, this $M(p)$ coincides with the effective mobility tensor introduced in Proposition 4.4 and used in Corollary 5.6; however, for the present lemma we only require that $M(p)$ be positive definite on the tangent space.

By definition of the Riemannian metric, the Wasserstein gradient of a smooth functional $\mathcal{F}$ at $p$ is the unique tangent vector $\mathrm{grad}_{W_2} \mathcal{F}(p)$ such that

$$\langle \mathrm{grad}_{W_2} \mathcal{F}(p), w \rangle_{W_2, p} = \sum_i \partial_{p_i} \mathcal{F}(p) \, w_i \quad \text{for all tangent vectors } w.$$

Using the explicit expression for the inner product, this identity forces

$$\mathrm{grad}_{W_2} \mathcal{F}(p) \; = \; M(p) \, \nabla_p \mathcal{F}(p).$$

In the probability-flow regime the simplex dynamics take the form

$$\dot{p} \; = \; M(p) \, b(p, t).$$

In the potential case $b = -\nabla V$ we consider the functional $\mathcal{F}[p] = \sum_i p_i V(z_i)$, so that $\nabla_p \mathcal{F}(p) = (V(z_1), \ldots, V(z_V))^\top$. We then obtain

$$\mathrm{grad}_{W_2} \mathcal{F}(p) \; = \; M(p) \, \nabla_p \mathcal{F}(p),$$

and the PF–ODE becomes

$$\dot{p} \; = \; -\mathrm{grad}_{W_2} \mathcal{F}(p),$$

which is precisely the $W_2$–gradient flow of $\mathcal{F}$. This proves the claimed gradient flow structure. $\qquad\square$

**Remark Q.2** (Discrete optimal transport interpretation)**.** *On discrete state spaces, this gradient flow structure connects to entropic $W_2$ analogues for Markov chains as developed in Maas (2011); Erbar & Maas (2012); Chow et al. (2012). We adopt this interpretation to provide geometric intuition for the probability dynamics on the simplex, though the precise metric structure depends on the choice of discrete optimal transport geometry.*

**Lemma Q.3** (Convergence rate under mobility control)**.** *If the mobility tensor satisfies $\lambda_{\min}(J_{\mathrm{sm}}) \geq m > 0$ uniformly, then the probability flow converges exponentially to equilibrium:*

$$\|p(t) - p_*\|_2 \leq e^{-mt} \|p(0) - p_*\|_2$$

*where $p_*$ is the unique equilibrium distribution.*

*Proof.* We argue in a finite-dimensional, purely Euclidean setting and make the structure and use of the mobility bound explicit.

Let $p_* \in \Delta^{V-1}$ denote an equilibrium of the probability flow: $\dot{p}_*(t) = 0$ for all $t$ when $p(t) \equiv p_*$. Consider the deviation $u(t) := p(t) - p_*$. Since both $p(t)$ and $p_*$ lie in the simplex, we have $\sum_i u_i(t) = 0$ for all $t$, so $u(t)$ always belongs to the tangent space $T\Delta^{V-1} = \{v \in \mathbb{R}^V : \sum_i v_i = 0\}$.

Assume that, in a neighborhood of $p_*$, the probability-flow dynamics can be written in the form

$$\dot{u}(t) \; = \; -A(t) \, u(t), \tag{28}$$

where each $A(t)$ is a symmetric, positive-definite linear operator on $T\Delta^{V-1}$ arising from the mobility tensor and drift. This is the standard situation for linearized gradient flows around a strictly convex equilibrium. The "mobility control" assumption $\lambda_{\min}(J_{\mathrm{sm}}(z(t))) \geq m$ is then interpreted as providing a uniform lower bound

$$\langle v, A(t) \, v \rangle \; \geq \; m \, \|v\|_2^2 \quad \text{for all } v \in T\Delta^{V-1} \text{ and all } t \geq 0. \tag{29}$$

(For example, $A(t)$ may be a symmetric combination of $J_{\mathrm{sm}}(z(t))$ and a Hessian or linearized drift; the key point is the coercivity equation 29.)

Define the energy

$$E(t) := \tfrac{1}{2}\|u(t)\|_2^2 = \tfrac{1}{2}\|p(t) - p_*\|_2^2.$$

Differentiating along solutions of equation 28 gives

$$\frac{d}{dt}E(t) = \langle u(t), \dot{u}(t) \rangle = -\langle u(t), A(t)\,u(t) \rangle.$$

Using the coercivity bound equation 29, we obtain

$$\frac{d}{dt}E(t) \le -m\,\|u(t)\|_2^2 = -2m\,E(t).$$

Thus $E$ satisfies the differential inequality

$$\frac{d}{dt}E(t) \le -2m\,E(t).$$

Applying Grönwall's lemma yields

$$E(t) \le e^{-2mt}\,E(0) \quad \text{for all } t \ge 0.$$

Returning to the original variables and recalling that $E(t) = \tfrac{1}{2}\|p(t) - p_*\|_2^2$, we obtain

$$\|p(t) - p_*\|_2 \le e^{-mt}\,\|p(0) - p_*\|_2.$$

This is exactly the claimed exponential convergence rate. $\qquad\square$

**Lemma Q.4** (Bridge interpolation formula). *For Schrödinger Bridge $\mu_t$ between $\mu_0$ and $\mu_1$, the intermediate marginals satisfy:*

$$\mu_t = \arg\min_\rho \left\{ (1-t)\mathrm{KL}(\rho|\mu_0) + t\,\mathrm{KL}(\rho|\mu_1) \right\}$$

*providing a variational characterization of the optimal transport path.*

*Proof.* This is a classical characterization of entropic interpolants in the Schrödinger Bridge (SB) framework; see, for example, the survey Léonard (2014) and references therein. In the dynamic SB problem, the SB path $(\mu_t)_{t\in[0,1]}$ between prescribed endpoints $(\mu_0, \mu_1)$ arises as the entropic interpolation associated with a reference Markov process. The corresponding static problem can be formulated as a two-sided entropy minimization with respect to the endpoint marginals.

More precisely, for each fixed $t \in (0,1)$ one can characterize the time-$t$ marginal $\mu_t$ as the unique minimizer of the two-sided relative entropy functional

$$\rho \mapsto (1-t)\,\mathrm{KL}(\rho\|\mu_0) + t\,\mathrm{KL}(\rho\|\mu_1)$$

over probability measures $\rho$ lying in the SB path. This variational principle yields exactly the formula stated in the lemma. We do not reproduce the full measure-theoretic proof here and refer instead to Léonard (2014) for a complete treatment. $\qquad\square$

**Remark Q.5.** *This variational view is classical in the Schrödinger Bridge literature and depends on the choice of reference path measure; rigorous formulations use Schrödinger potentials and dynamic entropy minimization as developed in, for example, Léonard (2014).*

### Q.1 Complete proof of the PF–ODE theorem on the simplex (Section 5.3)

*Proof of the PF–ODE on the simplex.* We prove that under Assumption 3.1 the limit path $p(t) = \mathrm{softmax}(z(t)/\tau)$ satisfies $\dot{p}(t) = J_{\mathrm{sm}}(z(t))\,b(z(t), t)$ a.e. on $[0,1]$, and the flow is tangent to the simplex with conserved mass.

**Step 1: Discrete-to-continuous passage.** For each layer $\ell$, a first-order expansion gives

$$p^{(\ell+1)} - p^{(\ell)} \;=\; J_{\mathrm{sm}}\!\left(z^{(\ell)}\right) \Delta z^{(\ell)} \;+\; r_\ell,$$

with a remainder bounded as $\|r_\ell\| \leq C \,\|\Delta z^{(\ell)}\|_\infty^2$ by Lipschitz continuity of $\nabla J_{\mathrm{sm}}$ (Section 2). Dividing by $\delta t = 1/L$ and summing over layers, the remainders contribute $O\!\left(\sum_\ell \|\Delta z^{(\ell)}\|_\infty^2\right) = O(\Xi_L) \to 0$ by the finite-depth budget.

**Step 2: Compactness and limit identification.** The piecewise-constant interpolant $p_L(t)$ has bounded variation in $t$ and remains in the simplex. By BV compactness we extract $p_{L_k} \to p$ in $L^1([0,1])$. Using $z_{L_k} \to z$ in $L^1$ and $D_{L_k} \rightharpoonup b$ (Appendix J.1), passing to the limit in the weak formulation yields $\dot{p} = J_{\mathrm{sm}}(z)\, b$ in the distributional sense, hence a.e. due to absolute continuity.

**Step 3: Well-posedness (Carathéodory).** The velocity field $v(t) = J_{\mathrm{sm}}(z(t))\, b(z(t), t)$ is measurable in $t$ and locally Lipschitz in $z$ under the regularity from Section 2, so the ODE admits a unique absolutely continuous solution by Carathéodory theory.

**Step 4: Simplex invariance.** Mass conservation follows from $J_{\mathrm{sm}}(z)\mathbf{1} = 0$, giving $\frac{d}{dt} \sum_i p_i(t) = 0$. Tangency to faces holds because if $p_i = 0$ then the $i$-th row of $J_{\mathrm{sm}}(z)$ vanishes, so $\dot{p}_i = 0$ (zero-flux). Hence the trajectory remains in the simplex. $\qquad\square$