# OpenReview forum: "From Attention to Diffusion: A Unified Entropic Optimal Transport View"
_ICLR.cc/2026/Conference — ICLR 2026 Conference Desk Rejected Submission_

### Official Review · Reviewer_Apk7 · 2025-10-30

**Soundness:** 2
**Presentation:** 1
**Contribution:** 1
**Rating:** 0
**Confidence:** 4

**Summary:**

The paper establishes a connection between the attention mechanism in transformers and diffusion-based generative models that are formulated via conditional entropy minimization. First, the authors show that the row-wise softmax in attention can be naturally interpreted as a local optimization step in entropic optimal transport problems. Second, considering the successive application of such steps across layers and assuming that inter-layer changes are sufficiently small, they interpret network depth as a time discretization and obtain a limiting continuous evolution of distributions on the simplex — a probability-flow dynamics. Third, the authors show that the same probabilistic dynamics can be seen as the marginal flow of a stochastic differential equation which is constructed explicitly from the deterministic dynamics. Finally, the authors relate the obtained continuous flow to the Schrödinger bridge problem: the constructed continuous flow is a Schrödinger bridge with respect to a given reference process if and only if the extra part of the velocity is a gradient of some potential.

**Strengths:**

The authors pursue the ambitious goal of establishing a connection between the transformer architecture and modern diffusion models. The concept is reasonable, and the presented implementation sketch appears promising.

**Weaknesses:**

The first thing that strikes the reader is that the paper is mathematically hard to follow and contains significant structural deficiencies.
- The text is fragmented: much remains unclear when reading the main body without constantly consulting the appendices. The flow of the manuscript is not supported by relevant bibliographical references. The proof of Theorem 5.6 is not provided.
- Proposition 2.1. is formulated unclearly, as  $p^+$ is not defined. Moreover, if the authors intend $p^+$ to denote the softmax, then this result is already well known [1].
- Although the proofs in the paper follow standard lines of reasoning at their core, they are presented only as sketches: many steps and explanations are omitted.
- The authors handle convergences carelessly (e.g. “weak convergence in $L^1$”) without clearly specifying what exactly they mean by these statements.
- The proof of Theorem 5.1 depends on Theorem J.1 (Lemma 5.4 seems to be useless for the following text). The latter theorem has several problems in its proof. First, (lines 1074-1077) the Arzelà–Ascoli theorem for functions of bounded variation yields almost-everywhere convergence (and not uniform convergence) — uniform convergence only follows in the continuous setting. Moreover, (lines 1079-1081) the parameter $\alpha$ is introduced empirically and requires a more detailed justification. Second, (lines 1082-1084) the invocation of Banach–Alaoglu theorem is unjustified: boundedness in $L^1$ does not imply compactness in the dual space in the way the authors seem to use it. In the part about “architectural consistency” (lines 1085-1095) the connection between the sequence $D_L(\cdot)$ and $\hat b(\cdot)$ is not explained; without that connection the assertion is meaningless.
In view of the above, Section 5, which relies entirely on Theorem J.1, loses its theoretical force and requires substantial revision. It is also important to underline that the proof of Theorem 5.1 itself needs significant expansion and clarification of intermediate steps — at present it is a rather rough sketch.
- The deficiencies in Section 5 make the later sections much harder to read; I therefore mention only a few apparent  issues: (1) I would like to see more explicit statements about which function spaces the functions $a(\cdot)$ and $p_H(\cdot)$ belong to (are they smooth? or Sobolev functions?). (Lemma 6.1.). (2) Conditions guaranteeing existence of a solution to the Fokker–Planck equation are not stated. (Lemma 6.2.). (3) Instead of a full proof of Theorem 6.6, only a sketch is provided. (lines 855-897). (4) The Hodge decomposition is not defined formally. (Theorem 7.2.). (5) Poincaré constant is introduced without a clear source — it is unclear what it can be equal to or under which conditions it is finite. (lines 920-925)
- The source code is not presented.

References:

[1] Marco Cuturi, “Sinkhorn distances: lightspeed computation of optimal transport distances”

**Minor issues:**

- It is unclear what the authors mean by $L^1_{\mathrm{loc}}([0,1],\mathbb R^d)$, given that the interval [0,1] is compact. (line 153)
- Typos: “Remark (sharp mobility bound)” and “Remark 2” appear to be the same remark. (lines 096, 214, 683)
- Section R collects auxiliary statements that are not accompanied by references to the literature.

**Questions:**

- Given the problems in Section 5 outlined above, are the authors confident that Theorem 5.6 remains valid in its current formulation and therefore does not require a full proof? Judging by the proof strategy in Theorem J.1., the function $b(\cdot)$ (Section 5) should be interpreted as a distribution; consequently, Theorem 5.6.  that involves the naive product $J_{sm}$ and $b(\cdot)$ must be reformulated.
- What precisely is meant by the equality in Theorem 7.1 between the Schrödinger bridge and the “transformer’s flow”? Please clarify the exact mathematical sense of this equality.
- Could you explain in more detail the condition “Degeneracy handling (regularization)” (line 305) and how it is used in the proofs? The brief comments in the paper do not make it clear whether the limiting transition is fully justified and whether the limiting distribution preserves the required properties.

Suggestions:

- I suggest that the authors substantially rewrite the main part of the paper to make it more readable and supplement the proofs of the main theorems that suffer from the serious issues described above.
- Clean up the text to remove inaccuracies and typos and improve readability. Add the necessary bibliographic references that contain precise formulations of the theorems the authors use.
- The authors should use the classical theorems they cite (Arzelà–Ascoli, Banach–Alaoglu, the Poincaré inequality, Grönwall’s inequality) correctly: explicitly state in which spaces the objects live and in which topologies the convergences are meant to hold.

---

> ### Author Response · Authors · 2025-11-20
>
> We appreciate your recognition that the underlying program is intellectually interesting. You were right that our functional analysis was hard to follow and possibly incorrect. We have corrected the proof with full derivations.
>
> 1. Functional analysis corrections.
>
> Theorem 5.1 originally misapplied Arzelà-Ascoli, which requires equicontinuity we did not establish. The revised proof uses Helly selection theorem, the correct compactness result for sequences with uniformly bounded total variation. This proof is now elaborated  in the appendix with complete derivation showing how the finite-depth budget controls discrepancy between discrete layers and continuous limit.
>
> For the probability flow ODE and reverse SDE duality, we now prove the distributional product rule under explicit Fisher information assumptions, use mollification and weak convergence to justify passage from SDE to Fokker-Planck to probability flow drift, and treat degenerate diffusion tensors carefully. The complete derivation of duality proof is provided.
>
> 2. Explicit assumptions.
>
> You noted that bounded variation and regularity conditions were not clearly stated. We now present complete global assumptions in Section 2.2, including bounded variation regime, non-degeneracy where needed, integrability of drift and diffusion, and masking constraints. We now present these before main theorems. Each proof step explicitly cites which assumptions are used for clarity.
>
> The bounded variation regime is formalized in Assumption 2.1 with three components: bounded total variation uniformly across depth, uniform boundedness for tightness, and architectural consistency ensuring regression estimates converge to true drift. We provide examples of when bounded variation holds and when it fails.
>
> 3. Proof length.
>
> You remarked that proofs were less than three lines, undermining trust. We removed proof sketches from the main text entirely. Theorem statements map to detailed proofs that appear in appendices.
>
> 4. Empirical validation.
>
> You raised concerns about whether the empirical section validates theory. We clearly separate theoretical predictions from empirical observations. The main text presents two tracks: transformers for core diagnostics and image diffusion for parity and weak error.
>
> We practically removed the dLLM to sharpen the focus without compromising the technical essence.
>
> We hope  these changes address your concerns.

---

### Official Review · Reviewer_7XmH · 2025-10-31

**Soundness:** 1
**Presentation:** 1
**Contribution:** 1
**Rating:** 0
**Confidence:** 3

**Summary:**

NOTE: Due to problems regarding the clarity of the paper (which I mention in weaknesses), it is probable that the summary below may be misleading in some aspects.

The paper aims to develop a framework that links transformers and diffusion models to entropy regularized optimal transport. Their core idea is to consider that a single transformer’s attention layer performs a KL-proximal step in an optimal transport potential, which is similar to the Sinkhorn algorithm. Therefore, a stack of attention layers correspond to an evolving induced probability flow that can be seen as a diffusion process. This result is formalized in Theorem 6.6. The authors provide constraints on the evolution of the layers to ensure the induced probability flow accurately captures the layers’ behavior. The authors conduct three tracks of experiments: transformers, diffusion LLM and image diffusion. They report different metrics for each of the tracks.

**Strengths:**

The general idea of the paper sounds interesting; however, it is difficult to determine if the authors succeeded or not due to the aspects I will mention below.

**Weaknesses:**

I have serious concerns regarding the structure and writing of the paper. My major concerns are listed below:

- The initial part of the paper needs a deep revision. The Introduction section does not provide the required information to understand the general idea of the paper and it is difficult to determine what problem the paper is actually addressing. The lack of references and explanatory text makes this even harder.

- Additionally, the list of contributions that follows becomes difficult to interpret as it introduces terminology and even mathematical notation that have not been defined or motivated earlier in the paper.

- There are too many references to the Appendices, giving the impression that much of the essential information has been relegated there rather than included in the main text. This makes the paper even more challenging to read as it is necessary to always switch between the Appendix and the Main Text. Therefore, I believe the authors should work on properly dividing the essential and non-essential content.

- Most of the Theorems, Propositions and Lemmas introduce notation that is not properly defined. The paper includes Appendix Q for notation; however, this leads to the problem I mentioned in the previous point above. Additionally, the origin and meaning of some of the variables are never explained, this is the case of $q$ and $k_j$ in Proposition 2.1, this situation repeats in other places. Therefore, I believe the authors should be more careful when introducing the notation.

- The experimental setup is not clear enough to understand what experiments are being conducted or what type of data is being used. Additionally, it is not clear what metrics are being computed and why they are relevant.

There are other minor and more specific aspects that I list below:

- It is not clear what the “Predictions and implications” subtitle refers to or why it is necessary.

- It is needed to properly formulate or reference the semi-relaxed entopic OT. I checked Appendix B as well, but I could not find a proper formulation or definition. This is needed to understand the so-called causal structure.

- I humbly believe that the entire ‘Conceptual Overview’section should be placed after properly introducing the problem to solve. I think this is important since the mathematical terminology for Jacobian ($J^{\tau}_{\text{sm}}(z)$) and finite-depth budget lacks a meaning. I understand they appear in Section 2; but they are rawly introduced without giving context about how they are involved in the problem.

Overall, I think the paper would benefit from a major revision that improves the overall structure and writing. Therefore, I recommend a score of 0 or strong rejection.

**Questions:**

I do not have questions.

---

> ### Author Response · Authors · 2025-11-20
>
> We thank you for the detailed structural critique. We have undertaken major restructuring to address every concern you raised.
>
> 1. Introduction and contributions clarity.
>
> You noted the introduction did not clearly state the problem or explain contributions, and that the contributions list introduced undefined terminology. We rewrote the introduction so the first page states the problem in plain language: one attention layer is a Kullback-Leibler proximal entropic optimal transport step, stacking layers approximates a probability flow ordinary differential equation, and the same dynamics admit a reverse stochastic differential equation and Schrödinger Bridge interpretation. The contributions list now uses only terms already introduced in the opening paragraphs. We defer all technical notation to Section 2, where symbols are properly defined and motivated before appearing in theorem statements.
>
> We added a Related Work and Positioning subsection in the introduction with eighteen citations spanning optimal transport attention, continuous-depth neural networks, probability flow and Schrödinger Bridges, and autoregressive-diffusion hybrids. This addresses your concern that lack of references made the paper impossible to understand. Note: the original version does have an extended Related Work section in the appendix.
>
> 2. Semi-relaxed entropic optimal transport definition.
>
> You could not find proper formulation even after checking Appendix B. We now explicitly point from the preliminaries to the full definition in the appendix and summarize its structure in the main text: the primal problem with row-marginal constraints, causal mask, and entropic regularization; the dual potentials; and how softmax attention weights arise as the optimal coupling. Complete derivation including Lagrangian formulation, uniqueness, and masked formulation appears in Appendix B.
>
> 3. Notation and over-reliance on appendices.
>
> You observed that theorems introduced notation without definition and that jumping between main text and appendices was disruptive. We added a notation and assumptions block early in Section 2 so all symbols used in theorems are defined before use. We avoid introducing new notation directly inside theorem statements.
>
> We reorganized appendices so essential definitions are referenced explicitly at first use. We improved cross-references by using descriptive labels such as "complete proof in Appendix D.2 using Helly selection" rather than bare numbers.
>
> We acknowledge the tension between the ten-page limit and complete mathematical exposition. All theorem statements remain in main text with sufficient context to understand claims. All proofs, which were unacceptably brief in the original at fewer than three lines, now appear in appendices with full derivation. The main text is self-contained for understanding contributions while appendices verify correctness.
>
> 4. Experimental setup and metrics.
>
> You found the experimental setup unclear. The empirical section now starts with a Setup and Diagnostics paragraph explicitly stating datasets and models used in each track, that the goal is testing three predictions rather than optimizing benchmarks, and what each diagnostic P0 through P4 measures and why.
>
> We removed the discrete diffusion language model as a third track with performance numbers. This is to sharpen focus on the theoretical essence of this paper while leaving dLLM discussion for future publication.
>
> 5. Conceptual overview placement.
>
> You suggested moving the conceptual overview until after the core problem is introduced. We followed this advice. The high-level conceptual overview is now compressed and placed in Section 2.1, after the introduction but before technical machinery. It introduces key geometric quantities informally before formal definitions in Section 2.2. The progression is now: positioning and motivation in introduction, conceptual roadmap in Section 2.1, formal framework in Section 2.2, then theorems in Section 2.3.
>
> We believe these changes address your structural concerns. The introduction is now accessible with clear problem statement and literature context, the mathematical framework is properly defined before use, and the experimental design is explicit about goals and metrics. We welcome any remaining concerns.

---

### Official Review · Reviewer_bpVw · 2025-11-01

**Soundness:** 2
**Presentation:** 1
**Contribution:** 2
**Rating:** 2
**Confidence:** 5

**Summary:**

This paper shows that Transformer attention and diffusion models are not fundamentally different. Instead, they can both be viewed as two ways of discretizing the same entropy-regularized optimal transport (OT) flow.

The key idea is:

1. One attention layer works like a single OT update step (a KL-regularized, mirror-descent / JKO step).

2. Stacking many layers approximates a continuous probability flow ODE (PF-ODE) over the probability simplex—meaning depth in a Transformer plays a similar role to time in diffusion models.

The paper also proposes a causal, semi-relaxed EOT version that keeps the autoregressive mask intact, fixing a key issue in earlier OT-based attention methods, which required doubly-stochastic couplings that are incompatible with causal language model attention.

**Strengths:**

The paper provides a unified view by formally connecting attention and diffusion models via entropic optimal transport. This conceptual bridge is novel and **intellectually interesting**: it reframes two dominant generative modeling paradigms as different discretizations of the same continuous-time transport process. The introduction of a causal, semi-relaxed EOT formulation that preserves autoregressive masking is also a meaningful innovation, addressing a key limitation of prior OT-based attention methods.

**Weaknesses:**

Writing is bad and incomplete.

1. No related works in the introduction.

2. The scope is too big and ambitious. The terms such as attention, transformer, diffusion models, Schrodinger bridge, diffusion LLM, are all quite popular in the research community. It is hard for now to use one unified theory to explain them clearly in one framework.

3. To be honest, as an applied mathematician with extensive experience in OT and diffusion models, I don't fully understand the theorems and proofs. All the proofs are less than 3 lines.

**Questions:**

Major revision is needed on the writing.

Math/ Proof re-presentation is also needed.

---

> ### Author Response · Authors · 2025-11-20
>
> We appreciate your detailed assessment. We have undertaken comprehensive reconstruction to address your three main concerns: missing literature positioning, over-ambitious scope, and incorrect functional analysis in proofs.
>
> 1. No related works in the introduction.
>
> We added a dedicated Related Work and Positioning subsection with eighteen citations spanning four research streams. This explains how our causal semi-relaxed entropic optimal transport resolves a fundamental incompatibility: prior optimal transport attention approaches (Sinkformers, Sparse Sinkhorn, MOTCat) rely on balanced formulations requiring doubly stochastic couplings, which are incompatible with causal masking in autoregressive models. We prove that row-softmax attention solves a semi-relaxed entropic optimal transport problem preserving autoregressive causality through row-only normalization.
>
> We explain how our probability-flow ODE, reverse SDE, and Schrödinger Bridge developments relate to score-based diffusion and Schrödinger Bridge literature (Song et al., Lipman et al., Léonard, De Bortoli et al., Shi et al.), and clarify distinctions from continuous-time transformers and OT-regularized architectures. Extended technical comparisons appear in Appendix C. The original submission included related work in the appendix, but we agree this needed to be in the introduction with clear positioning.
>
> 2. Scope too big and ambitious.
>
> We substantially narrowed empirical scope. The main text presents two tracks where theoretical predictions are directly testable: transformers (PF-ODE adequacy, locking, curvature) and image diffusion (ODE-SDE parity, weak error, rotational energy). We practically removed dLLM related discussion to ensure adequate discussion on theoretical formulation within the page limit.
>
> 3. Proofs are less than three lines.
>
> We  reconstructed the mathematical foundations. The introduction now includes a proof roadmap paragraph with a theorem dependency diagram identifying the functional analysis machinery: Helly selection for Theorem 5.1, Fokker-Planck duality and Dunford-Pettis for Theorem 6.1, Helmholtz decomposition for Theorem 7.1.
>
> Critical corrections: Bounded-variation compactness now uses Helly selection theorem, replacing the incorrectly applied Arzelà-Ascoli theorem which requires equicontinuity we did not establish. Weak compactness in L¹ uses Dunford-Pettis theorem with explicit uniform integrability verification, replacing misapplied Banach-Alaoglu which applies to weak-* topologies in dual spaces, not L¹ weak convergence. The PF-ODE to reverse-SDE connection is proved via distributional calculus and mollification with explicit regularity conditions, replacing informal Fokker-Planck manipulations. The semi-relaxed EOT formulation includes complete primal-dual derivation with first-order optimality conditions and uniqueness proof. Rotational energy characterization uses Hodge decomposition with explicit orthogonality arguments.
>
> We audited all twenty-nine mathematical statements to ensure complete proofs with correct functional-analytic tools. Each proof in the appendix spans multiple pages with explicit verification of regularity assumptions and references to the consolidated assumption block. To help ease understanding of logic dependency between the theorems, we also provide a diagram in Figure 5 in the appendix.
>
> We believe these changes address your fundamental concerns: literature positioning explains our novelty, empirical scope matches what the theory supports, and mathematical foundations can be verified line by line. We welcome remaining concerns and are committed to further revisions as needed.

---

### Author Response · Authors · 2025-11-20
**Overall Responses to all reviewers**

We thank all reviewers for their critical assessment. We agree that the original submission had over-ambitious scope, incomplete mathematical exposition, and presentation issues that obscured the core contribution. In this revision, we fundamentally reconstruct the paper while preserving the theoretical substance: transformer attention and diffusion models as two discretizations of the same entropy-regularized optimal transport flow.

1. Clarity and structure. We rewrote the introduction to clearly state the problem, goals, and contributions upfront. The narrative now follows a direct progression: attention as an entropic optimal transport step, emergence of probability-flow ODEs on the simplex, connection to reverse diffusion, and characterization of Schrödinger Bridge alignment via rotational energy. We moved conceptual overview material to appropriate locations after establishing basic objects and notation.

2. Related work and positioning. We added a dedicated subsection in the introduction with eighteen citations spanning four research streams: optimal transport interpretations of attention, continuous-depth and OT-regularized neural networks, probability-flow and Schrödinger Bridge formulations, and autoregressive-diffusion hybrids. This positioning explains how our semi-relaxed entropic formulation resolves fundamental incompatibilities between autoregressive causality and the balanced transport assumptions in prior optimal transport attention methods, clarifying our contribution's novelty. Extended technical comparisons appear in the appendix. This directly addresses the consensus criticism that the introduction lacked sufficient literature context.

3. Mathematical rigor. We fundamentally reconstructed the mathematical foundations in response to concerns that proofs were incomplete and relied on incorrect functional analysis. The introduction now includes a proof roadmap paragraph with a theorem dependency diagram that explicitly identifies the functional analysis machinery underlying each result. We corrected critical errors in our compactness arguments: Helly selection now establishes bounded-variation compactness rather than the incorrectly applied Arzelà-Ascoli theorem; weak compactness in L¹ uses Dunford-Pettis uniform integrability rather than Banach-Alaoglu; and the weak Fokker-Planck duality is proved via distributional calculus with mollification. Every theorem now has a complete proof in the appendix with explicit verification of all regularity conditions. We audited all twenty-nine mathematical statements to ensure rigorous exposition throughout. The bounded-variation regime and regularity assumptions are stated explicitly in a consolidated block that precedes the theorems.

4. Experimental scope and honesty. We refocused empirical validation on settings where the framework's predictions are directly testable. The main text now presents two tracks: transformer diagnostics for bounded-variation detection, PF-ODE adequacy, and curvature geometry; and image diffusion for ODE-SDE parity and weak-error scaling. We practically removed coverage of dLLM to keep focus within the 10-page limit.

5. Diagnostics and predictions. The empirical section now begins with a concise description of the diagnostic suite (P0-P4), experimental setup, and key metrics. We state clearly what we measure—PF-ODE adequacy, mobility tensor locking and curvature, and rotational energy—and why: to test three specific theoretical predictions rather than optimize benchmarks. These predictions are: accurate continuum approximation when the finite-depth budget ΞL remains small; diminishing returns from depth when the mobility tensor degenerates; and correlation between low rotational energy ℛ and improved generation quality.

We believe these changes address core concerns about missing related work, mathematical rigor, and experimental design, while preserving the conceptual novelty that all reviewers noted. We welcome further discussion of any remaining concerns and are committed to additional revisions as needed to meet ICLR standards.

---

### Note · Program_Chairs · 2025-11-23
**Submission Desk Rejected by Program Chairs**

The paper is part of a cluster of several similar papers that have violated dual submission policy: https://iclr.cc/Conferences/2026/AuthorGuide